# Understanding Transferable Representation Learning and Zero-shot Transfer in CLIP

**Zixiang Chen**[‡*], **Yihe Deng**[‡*], **Yuanzhi Li**[◇], **Quanquan Gu**[‡]
[‡]Department of Computer Science, University of California, Los Angeles
[◇]Machine Learning Department, Carnegie Mellon University, Pittsburgh
{chenzx19, yihedeng}@cs.ucla.edu
yuanzhil@andrew.cmu.edu, qgu@cs.ucla.edu

## Abstract

Multi-modal learning has become increasingly popular due to its ability to leverage information from different data sources (e.g., text and images) to improve the model performance. Recently, CLIP has emerged as an effective approach that employs vision-language contrastive pretraining to learn joint image and text representations and exhibits remarkable performance in zero-shot learning and text-guided natural image generation. Despite the huge practical success of CLIP, its theoretical understanding remains elusive. In this paper, we formally study transferrable representation learning underlying CLIP and demonstrate how features from different modalities get aligned. We also analyze its zero-shot transfer performance on the downstream tasks. Inspired by our analysis, we propose a new CLIP-type approach, which achieves better performance than CLIP and other state-of-the-art methods on benchmark datasets.

## 1 Introduction

Multi-modal learning (Ngiam et al., 2011) integrates information from a variety of data types, resulting in AI systems that are both robust and precise. Recently, CLIP (Radford et al., 2021) emerged as a milestone work that leverages vision-language contrastive pretraining to jointly learn image and text embeddings, using the vast amounts of image-text data available on the web. During the training process, CLIP considers image-text data that appear together as positive pairs and other combinations as negative pairs. The goal is to maximize the embedding similarity for the positive pairs while minimizing it for the negative pairs. Remarkably, this approach has achieved significant success in zero-shot transfer (Lei Ba et al., 2015), indicating the model's ability to handle a great variety of tasks without prior exposure to any of their training data. Inspired by CLIP's groundbreaking zero-shot capabilities, subsequent studies (Yao et al., 2022; Li et al., 2022; Mu et al., 2022; Goel et al., 2022; Zhai et al., 2022; Alayrac et al., 2022) emerged with the primary objective of further enhancing CLIP's zero-shot performance. Despite the empirical success of CLIP in zero-shot transfer, the theoretical understanding of how it works remains elusive. An intriguing inquiry is thus: *How* does CLIP learn representations that are transferable to the various downstream tasks?

This paper delves into the mechanisms through which CLIP learns transferable representations (i.e., embeddings) and demonstrates how such representations ensure successful zero-shot transfer for downstream tasks. We begin with identifying several challenges associated with the theoretical analysis of the transfer mechanism in CLIP: (1) alignment between different modalities, (2) unique features in different feature domains, and (3) sparsity of shared features across domains. In particular, unlike unimodal contrastive learning where the embedding function is shared, CLIP employs different embedding functions $f$ and $g$ for different modalities. This difference poses the alignment challenge specific to multi-modal learning. Secondly, the feature domains lie in different spaces and may lack a one-to-one mapping. Some features are shared, while others are unique. Take Figure 1 as an example. The attribute "stop sign" is a shared feature in both the image and the text. However, the "blue sky" and "white cloud" are examples of unique features in the images that are not evident in the caption. This misalignment causes bad alignment at initialization. Lastly, the shared features in multi-modal contrastive learning (e.g., objects) can be sparse, compared to the unique features (e.g., textures, colors). Consequently, certain image-text combinations, despite not being paired, may still have

---

[*]Equal contribution.

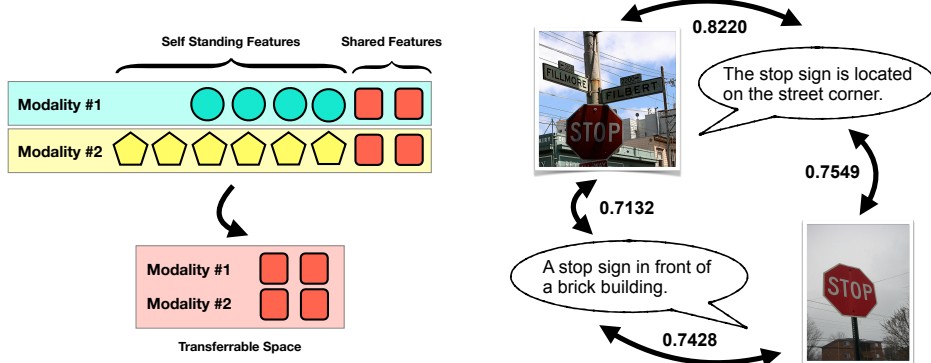

**Figure 1: Illustration of the Challenges.** Left: The feature domains are different and not one-to-one mapping. We need to learn transferable features while preserving the shared features. Right: The image-text data show in the same batch can have similar shared features since the shared features are sparse (here is "stop sign"). The learned similarities between each image-text pair are very close.

shared features, suggesting they should be treated as positive pairs. This challenges the traditional view of considering image-text data not paired together as negative pairs.

To tackle the above challenges, we present our theoretical result for transferable representation learning in CLIP and summarize our contributions as follows.

- We theoretically examine transferable representation learning in CLIP. Our analysis shows that if a near-optimal network is obtained on the training data, features from different modalities become aligned, enabling zero-shot learning if appropriate prompts are issued. We also demonstrate that, interestingly, contrastive learning with sparse features may lead to unexpected positive pairs. Therefore, we need to take it into careful consideration. Moreover, while previous studies typically require a very large batch size for training, our theoretical framework applies to small batches.
- Building upon our general theoretical findings, we delve deeper into specific cases, providing more comprehensive theoretical insights. We illustrate how multi-modal learning aligns different features and reveal when the learned features obtained by CLIP can outperform those obtained through naive square loss. By comparing CLIP loss and square loss, we formally established that CLIP is an effective learning objective for zero-shot transfer tasks, whereas square loss does not.
- We conduct experiments on real data to confirm our theoretical predictions. Furthermore, inspired by our theoretical findings, we propose a new regularization technique for CLIP that effectively leads to improved zero-shot performance. Empirical results confirm that the proposed regularization can effectively improve the zero-shot performance across various tasks.

**Notation.** We use lowercase letters, lowercase boldface letters, and uppercase boldface letters to denote scalars, vectors, and matrices, respectively. For a vector $\mathbf{x}$, we use $\|\mathbf{x}\|_2$ to denote its Euclidean norm. For a matrix $\mathbf{W}$, we use $\|\mathbf{W}\|_F$ to denote its Frobenius norm. Given two sequences $\{x_n\}$ and $\{y_n\}$, we denote $x_n = \mathcal{O}(y_n)$ if $|x_n| \leq C_1|y_n|$ for some absolute positive constant $C_1$, $x_n = \Omega(y_n)$ if $|x_n| \geq C_2|y_n|$ for some absolute positive constant $C_2$, and $x_n = \Theta(y_n)$ if $C_3|y_n| \leq |x_n| \leq C_4|y_n|$ for some absolute constants $C_3, C_4 > 0$. We also use $\widetilde{\mathcal{O}}(\cdot)$ to hide logarithmic factors of $d$ in $\mathcal{O}(\cdot)$. Additionally, we denote $x_n = \text{poly}(y_n)$ if $x_n = \mathcal{O}(y_n^D)$ for some positive constant $D$, and $x_n = \text{polylog}(y_n)$ if $x_n = \text{poly}(\log(y_n))$. We also denote by $x_n = o(y_n)$ if $\lim_{n \to \infty} x_n/y_n = 0$. Finally we use $[N]$ to denote the index set $\{1, \ldots, N\}$. In the function space, let $B_r(f)$ denote the ball of radius $r$ centered at $f$, with the metrics $\|\cdot\|_\infty$. A set $C$ is the covering of function class $\mathcal{F}$ with radius $r$, if and only if $\mathcal{F} \subseteq \cup_{f \in C} B_r(f)$. The covering number of $\mathcal{F}$ with radius $r$ is the minimum cardinality of any covering of $\mathcal{F}$, denoted as $\mathcal{N}(\mathcal{F}, r)$.

## 2 RELATED WORK

**Vision-Language Pre-Training.** While labeled data are expensive and relatively scarce, images paired with text descriptions are available in much larger volumes (Thomee et al., 2016). Consequently, numerous studies (Gomez et al., 2017; Sariyildiz et al., 2020; Desai & Johnson, 2021; Zhang et al., 2022; Liang et al., 2023) have focused on leveraging free-form natural language supervision to learn visual representations. Recently, CLIP (Radford et al., 2021) and ALIGN (Jia et al., 2021) have emerged as prominent works extending contrastive learning to the vision-language pre-training framework. Built upon CLIP's success, several studies (Pham et al., 2021; Gao et al., 2022; Saito et al., 2022) have refined CLIP's contrastive methodology to better learn from web-scale image-text data. Notably, UniCL (Yang et al., 2022) additionally incorporates image-label data, enabling the

identification of a broader range of positive pairs. FILIP (Yao et al., 2022) introduces a fine-grained contrastive loss tailored for transformer architectures. DeCLIP (Li et al., 2022) and SLIP (Mu et al., 2022) additionally incorporate single-modality self-supervised learning. CyCLIP (Goel et al., 2022) introduces two regularizing terms enforcing cross-modal and in-modal consistency. LiT (Zhai et al., 2022) and Flamingo (Alayrac et al., 2022) consider training from pre-trained single-modality models. In our empirical validation of theoretical findings, we employ the same setting and train from pre-trained image and text encoders.

**Theory of self-supervised learning.** Numerous studies have been conducted to understand *unimodal* contrastive learning, a widely used self-supervised learning approach rooted in data augmentation (Saunshi et al., 2019; Tsai et al., 2020; Mitrovic et al., 2020; Tian et al., 2020; Wang & Isola, 2020; Chen et al., 2021; Wang & Liu, 2021; Tosh et al., 2021b;a; HaoChen et al., 2021; Wen & Li, 2021; Saunshi et al., 2022). In multimodal learning, theoretical explanation has been explored in several studies (Zadeh et al., 2020; Huang et al., 2021; Lee et al., 2020; Nakada et al., 2023). These works have established that multimodal learning can surpass unimodal learning in terms of performance. For instance, Lee et al. (2020) employed square loss prediction to learn image representations under certain conditional independence assumptions, offering generalization performance guarantees. Meanwhile, Nakada et al. (2023) examined CLIP within specific linear representation settings and emphasized its correlation with singular value decomposition (SVD). We note that, these related works have not considered the zero-shot transfer mechanism and thus can't adequately explain the zero-shot transfer capability of CLIP.

## 3 PROBLEM SETTING AND PRELIMINARIES

### 3.1 DATA DISTRIBUTION

In our paper, we focus on the setting where the image $\mathbf{x}$ and the text $\mathbf{y}$ are conditionally independent given the shared feature $\mathbf{z}$.

**Assumption 3.1.** Let $(\mathbf{x}, \mathbf{y})$ be generated from the joint distribution $\mathcal{D}_{\mathbf{x} \times \mathbf{y}}$. We assume $\mathbf{z}$ to be a shared feature of $\mathbf{x}, \mathbf{y}$ satisfying $\mathbf{x} \perp \mathbf{y} | \mathbf{z}$, and further denote $(\mathbf{x}, \mathbf{y}, \mathbf{z})$ that follows the joint distribution $\mathcal{D}_{\mathbf{x} \times \mathbf{y} \times \mathbf{z}}$ with marginal distributions $\mathcal{D}_{\mathbf{x} \times \mathbf{z}}, \mathcal{D}_{\mathbf{y} \times \mathbf{z}}$. We further assume $\mathbf{z}$ to be a discrete and sparse random variable $\mathbf{z} \in \mathcal{V} = \{\mathbf{v}_1, \ldots, \mathbf{v}_K\}$ with $p_k := \mathbb{P}(\mathbf{z} = \mathbf{v}_k)$.

Intuitively speaking, the shared feature $\mathbf{z}$ in the above assumption may denote a set of shared topics or keywords underlying image $\mathbf{x}$ and text $\mathbf{y}$. We can consider the following simple example to understand it. Let $\mathbf{z} = [0, 1, 0, 1]^\top$ represent the existence of topics "chair" and "table" and the absence of topics "car" and "train". Then, $\mathbf{x}$ and $\mathbf{y}$ are generated given $\mathbf{z}$ such that they both include "chair" and "table", yet with different unique features and noises.

**Remark 3.2.** The assumption of conditional independence is frequently made in the analysis of self-supervised learning (Saunshi et al., 2019; Lee et al., 2021) and dimension reduction algorithms (Fukumizu et al., 2004; 2009). Under the premise that $\mathbf{x}, \mathbf{y}$ are conditionally independent (CI) given $\mathbf{z}$, it can be posited that any additional patterns found within $\mathbf{x} | \mathbf{z}$ and $\mathbf{y} | \mathbf{z}$ should be interpreted as unique features. Notably, in the absence of discrete and sparse constraints, a suitable $\mathbf{z}$ can always be found, given that one could simply assign $\mathbf{z} = \mathbf{x}$ or $\mathbf{z} = \mathbf{y}$. From the generative model's point of view, Assumption 3.1 naively holds when the data are from some generator with $\mathbf{x} = T_1(\mathbf{z}, \boldsymbol{\xi})$ and $\mathbf{y} = T_2(\mathbf{z}, \boldsymbol{\zeta})$ where $\boldsymbol{\xi} \perp \boldsymbol{\zeta} | \mathbf{z}$.

### 3.2 LEARNING VIA CONTRASTIVE LOSS

CLIP is trained on millions of image and text pairs. Formally, we assume the data set $S$ is drawn from the distribution $\mathcal{D}_{\mathbf{x} \times \mathbf{y}}$ defined in Assumption 3.1. The CLIP architecture has three main components: (i) an image encoder network $\mathbf{g}$ that can encode the image $\mathbf{x}$ into the embedding $\mathbf{g}(\mathbf{x}) \in \mathbb{R}^d$; (ii) a text encoder network $\mathbf{h}$ that can encode the text $\mathbf{y}$ into an embedding vector $\mathbf{h}(\mathbf{y}) \in \mathbb{R}^d$; and (iii) a score function $f(\mathbf{x}, \mathbf{y}) = \mathbf{sim}(\mathbf{g}, \mathbf{h})$ that measures the similarity between the image $\mathbf{x}$ and the text $\mathbf{y}$ given their embeddings $\mathbf{g}, \mathbf{h}$ $\left( \text{e.g.,} f(\mathbf{x}, \mathbf{y}) = \langle \mathbf{g}(\mathbf{x}), \mathbf{h}(\mathbf{y}) \rangle \right)$.

During the training, we will sample a batch of image-captions pairs $S' = \{\mathbf{x}_i, \mathbf{y}_i\}_{i=1}^B \subseteq S$. The contrastive objective in CLIP aims to align the image representation $\mathbf{g}(\mathbf{x})$ and text representations $\mathbf{h}(\mathbf{y})$ by minimizing the following loss function,

$$
\begin{aligned}
L_{S'}(f, \tau) &= \frac{1}{B} \sum_{i \in S'} -\log \left( \frac{\exp\left(f(\mathbf{x}_i, \mathbf{y}_i)/\tau\right)}{\sum_{j \in S'} \exp\left(f(\mathbf{x}_j, \mathbf{y}_i)/\tau\right)} \right) + \frac{1}{B} \sum_{i \in S'} -\log \left( \frac{\exp\left(f(\mathbf{x}_i, \mathbf{y}_i)/\tau\right)}{\sum_{j \in S'} \exp\left(f(\mathbf{x}_i, \mathbf{y}_j)/\tau\right)} \right) \\
&= \frac{1}{B} \sum_{i \in S'} \log \left( \sum_{j \in S'} \exp\left(\left[f(\mathbf{x}_j, \mathbf{y}_i) - f(\mathbf{x}_i, \mathbf{y}_i)\right]/\tau\right) \right)
\end{aligned}
$$

$$+ \frac{1}{B} \sum_{i \in S'} \log \left( \sum_{j \in S'} \exp \left( [f(\mathbf{x}_i, \mathbf{y}_j) - f(\mathbf{x}_i, \mathbf{y}_i)] / \tau \right) \right), \tag{3.1}$$

where $\tau > 0$ is a temperature parameter. The training loss $L_{S'}$ over a single epoch can be viewed as the empirical version of the following population loss,

$$L_{\mathcal{D}^B}(f, \tau) = \mathbb{E} \left[ \log \left( \sum_{t \in [B]} \exp \left( [f(\mathbf{x}_1, \mathbf{y}_t) - f(\mathbf{x}_1, \mathbf{y}_1)] / \tau \right) \right) \right]$$
$$+ \mathbb{E} \left[ \log \left( \sum_{t \in [B]} \exp \left( [f(\mathbf{x}_t, \mathbf{y}_1) - f(\mathbf{x}_1, \mathbf{y}_1)] / \tau \right) \right) \right], \tag{3.2}$$

where the expectation is taken with respect to all $B$ random pairs $(\mathbf{x}_t, \mathbf{y}_t)$ i.i.d. sampled from $\mathcal{D}_{\mathbf{x} \times \mathbf{y}}$. Therefore, CLIP learns the score function $f$ with the corresponding representations $\mathbf{g}$ and $\mathbf{h}$ by minimizing $L_{\mathcal{D}^B}(f, \tau)$. In fact, we can divide the training dataset $S$ into $n$ batches $\cup_{k \in [n]} \mathcal{S}_k$. The following theorem shows that the empirical loss $\widehat{\mathbb{E}}_S(f, \tau) := (1/n) \sum_{k \in [n]} L_{S_k}(f, \tau)$ concentrates on the population loss when $n$ is large enough.

**Theorem 3.3.** Suppose $\delta \in (0, 1)$ and $n \geq (8\tau^{-1}\epsilon^{-2}M \log B) \log(2\mathcal{N}(\mathcal{F}, \epsilon/8M)/\delta)$, then with probability at least $1 - \delta$, we have

$$|\widehat{L}_S(f, \tau) - L_{\mathcal{D}^B}(f, \tau)| \leq \epsilon$$

for all function $f \in \mathcal{F}$ and $|f| \leq M$, where $\mathcal{N}(\mathcal{F}, \epsilon)$ is the covering number of $\mathcal{F}$.

Theorem 3.3 shows that the generalization gap $|\widehat{L}_S(f, \tau) - L_{\mathcal{D}^B}(f, \tau)|$ approaches zero as the number of batches $n$ increase. In practice, the batch size is limited by the GPU's memory and is smaller than the number of batches (or the number of training examples). Therefore, instead of letting the batch size $B$ go to infinity like in prior studies (Wang & Isola, 2020; Pham et al., 2021), we keep the batch size $B$ as a constant in (3.2) and Theorem 3.3 to enable the analysis of CLIP even for small batches. Pham et al. (2021) also provided the generalization gap for CLIP. However, their result is for $B \to \infty$ and a loss function without the log term, i.e., $\exp \left( f(\mathbf{x}_i, \mathbf{y}_i) / \tau \right) / \left( \sum_{j \in S'} \exp \left( f(\mathbf{x}_j, \mathbf{y}_i) / \tau \right) \right)$.

## 4 TRANSFERRABLE REPRESENTATION LEARNING

The key idea of CLIP is to pull the embeddings of positive image-text pairs together while pushing the embeddings of negative pairs apart. For the data pair $(\mathbf{x}, \mathbf{y}')$ generated with $\mathbf{x} \sim \mathcal{D}_{\mathbf{x}|\mathbf{z}}, \mathbf{y}' \sim \mathcal{D}_{\mathbf{y}|\mathbf{z}'}$, $(\mathbf{x}, \mathbf{y}')$ is a positive pair if $\mathbf{z} = \mathbf{z}'$ and a negative pair if $\mathbf{z} \neq \mathbf{z}'$. The reason is that when $\mathbf{z} = \mathbf{z}'$, the joint distribution of $(\mathbf{x}, \mathbf{y}')$ is the same as the joint distribution of $(\mathbf{x}, \mathbf{y}) \sim \mathcal{D}_{\mathbf{x} \times \mathbf{y}|\mathbf{z}}$ since $\mathbf{x}, \mathbf{y}$ are mutually independent given the latent variable $\mathbf{z}$. Next, we will show that the learning objective (3.2) will lead to the distinguishable representation of different latent variables $\mathbf{z}$ under certain assumptions.

**Assumption 4.1 ($(\alpha, \beta, \gamma)$-Completeness).** There exists a score function $f^*$ bounded by 1 (i.e., $|f^*| \leq 1$) with $f^* = \mathbf{sim}(\mathbf{g}^*, \mathbf{h}^*)$ satisfying the following properties,
- For any $\mathbf{z} \neq \mathbf{z}'$, let $\mathbf{x} \sim \mathcal{D}_{\mathbf{x}|\mathbf{z}}, \mathbf{y} \sim \mathcal{D}_{\mathbf{y}|\mathbf{z}}, \mathbf{x}' \sim \mathcal{D}_{\mathbf{x}'|\mathbf{z}'}, \mathbf{y}' \sim \mathcal{D}_{\mathbf{y}'|\mathbf{z}'}$. With probability at least $1 - \alpha$, we have $f^*(\mathbf{x}', \mathbf{y}) \leq f^*(\mathbf{x}, \mathbf{y}) - \gamma$ and $f^*(\mathbf{x}, \mathbf{y}') \leq f^*(\mathbf{x}, \mathbf{y}) - \gamma$.
- Let $(\mathbf{x}, \mathbf{y}, \mathbf{z}) \sim \mathcal{D}_{\mathbf{x} \times \mathbf{y} \times \mathbf{z}}$, assume $\mathbb{E}_{(\mathbf{y}, \mathbf{z})} \left[ \mathrm{Var}_{\mathbf{x}|\mathbf{z}}(f^*(\mathbf{x}, \mathbf{y})) \right], \mathbb{E}_{(\mathbf{x}, \mathbf{z})} \left[ \mathrm{Var}_{\mathbf{y}|\mathbf{z}}(f^*(\mathbf{x}, \mathbf{y})) \right] \leq \beta$.

In simple terms, Assumption 4.1 is made on the data distribution to allow the *existence* of good encoding functions $\mathbf{g}^*$ and $\mathbf{h}^*$. Specifically, the first bullet guarantees that the data with different $\mathbf{z}$, the underlying shared feature, is well distinguishable with margin $\gamma$. If the data from different $\mathbf{z}$ does not satisfy this condition, the majority of the diagonal term $f(\mathbf{x}_i, \mathbf{y}_i)$ in (3.1) can be smaller than the off-diagonal term $f(\mathbf{x}_j, \mathbf{y}_i)$. In other words, all encoding functions may yield higher similarity score for negative pairs than positive pairs, which is not favored by the mechanism of CLIP. The second bullet requires the similarity score within each underlying shared feature not vary too much, which is naturally satisfied if the learned embeddings $\mathbf{g}(\mathbf{x}), \mathbf{h}(\mathbf{y})$ are consistent and do not vary too much given the same $\mathbf{z}$. In the following theorem, we establish the result that a CLIP model trained to convergence exhibits desirable properties in representation learning.

**Theorem 4.2.** Suppose Assumption 4.1 hold and we can find an $\epsilon$ approximate minimum $\widehat{f} \in \mathcal{F}$ with respect to the temperature $\tau$ such that $\widehat{f}$ is bounded by $M$ and

$$L_{\mathcal{D}^B}(\widehat{f}, \tau) \leq L_{\mathcal{D}^B}(f^*, \tau) + \epsilon. \tag{4.1}$$

Then the following results hold:

1. For $(\mathbf{x}, \mathbf{z}) \sim \mathcal{D}_{\mathbf{x} \times \mathbf{z}}$, $\{\mathbf{y}_k \sim \mathcal{D}_{\mathbf{y}|\mathbf{v}_k}, k \in [K]\}$, let $\mathbf{y}^* = \sum_{k \in [K]} \mathbb{1}(\mathbf{z} = \mathbf{v}_k)\mathbf{y}_k$, we have

$$\mathbb{E}\left[\log\left(\sum_{k \in [K]} \exp\left([\widehat{f}(\mathbf{x}, \mathbf{y}_k) - \widehat{f}(\mathbf{x}, \mathbf{y}^*)]/\tau\right)\right)\right] \leq \epsilon'. \tag{4.2}$$

2. For $(\mathbf{y}, \mathbf{z}) \sim \mathcal{D}_{\mathbf{y} \times \mathbf{z}}$, $\{\mathbf{x}_k \sim \mathcal{D}_{\mathbf{x}|\mathbf{v}_k}, k \in [K]\}$, let $\mathbf{x}^* = \sum_{k \in [K]} \mathbb{1}(\mathbf{z} = \mathbf{v}_k)\mathbf{x}_k$, we have

$$\mathbb{E}\left[\log\left(\sum_{k \in [K]} \exp\left([\widehat{f}(\mathbf{x}_k, \mathbf{y}) - \widehat{f}(\mathbf{x}^*, \mathbf{y})]/\tau\right)\right)\right] \leq \epsilon'. \tag{4.3}$$

3. For $(\mathbf{x}, \mathbf{y}, \mathbf{z}) \sim \mathcal{D}_{\mathbf{x} \times \mathbf{y} \times \mathbf{z}}$, variance $\mathbb{E}_{(\mathbf{y}, \mathbf{z})}\left[\text{Var}_{\mathbf{x}|\mathbf{z}}(\widehat{f}(\mathbf{x}, \mathbf{y}))\right] + \mathbb{E}_{(\mathbf{x}, \mathbf{z})}\left[\text{Var}_{\mathbf{y}|\mathbf{z}}(\widehat{f}(\mathbf{x}, \mathbf{y}))\right] \leq 16M^2\epsilon'$.

where $\epsilon' = (C_B + 2) \cdot [\epsilon + C\tau^{-1}MB\alpha + C\tau^{-1}(\beta MB)^{1/3} + 2B\exp(-\gamma/\tau)]$ and $C = \widetilde{O}(1)$, $C_B = \widetilde{O}(\max_k p_k^{-1}/B)$.

**Remark 4.3.** Theorem 4.2 establishes a soft margin between CLIP's learned embeddings on data of different $\mathbf{z}$'s. For instance, if an image $\mathbf{x}$ has a shared feature $\mathbf{z} = \mathbf{v}_1$, we have its accurate description $\mathbf{y}^* = \sum_{k \in [K]} \mathbb{1}(\mathbf{z} = \mathbf{v}_k)\mathbf{y}_k = \mathbf{y}_1$. From (4.2), it follows that $\log\left(\sum_{k \in [K]} \exp\left([\widehat{f}(\mathbf{x}, \mathbf{y}_k) - \widehat{f}(\mathbf{x}, \mathbf{y}_1)]/\tau\right)\right)$ is small. This can only occur when $\widehat{f}(\mathbf{x}, \mathbf{y}_k) < \widehat{f}(\mathbf{x}, \mathbf{y}_1)$ for all $k \geq 2$, i.e., the trained model always yield higher similarity score for this image-text pair as compared to all other texts generated on different topics. This outcome aligns with the expectation that image-text pairs with the same shared feature will yield the highest similarity score.

**Remark 4.4** (Choice of temperature parameter). When the data is well separated (i.e., $\alpha, \beta = 0$), a smaller temperature will invariably lead to a smaller $\epsilon'$ and, consequently, better performance. In practice, $\tau$ is typically set to be 0.01, a sufficiently small value that ensures the term $\exp(-\gamma/\tau)$ is less than 0.0000454 for $\gamma = 0.1$. However, when the data is nonseparable (i.e., $\alpha$ and $\beta$ exceed 0), a balance must be struck between the terms related to $\tau$. As a consequence, $\tau$ should not be too small. A reasonable choice would be $\tau = O(\gamma/\log(B/\epsilon))$.

**Remark 4.5** (Batch size). While we do not demand an increasing batch size $B$, our analysis does suggest a preference for larger batch sizes, as they can reduce the constant $C_B$ and consequently $\epsilon'$.

## 5 ZERO-SHOT TRANSFER

In this section, we will discuss why the embeddings learned by CLIP in Section 4 enable zero-shot transfer learning tasks. In the zero-shot transfer task, we have $K$ prompts $\{\mathbf{y}_k, k \in [K]\}$ where $\mathbf{y}_k \sim \mathcal{D}_{\mathbf{y}|\mathbf{v}_k}$. For a new image $\mathbf{x}$ generated from $\mathcal{D}_{\mathbf{x}}$, we want to predict the label of the shared feature $\mathbf{z}$ in $\mathbf{x}$. For example, if $\mathbf{x}$ has shared feature $\mathbf{v}_1$, then the label of $\mathbf{x}$ should be 1. As suggested by Radford et al. (2021), we calculate the similarity score between $\mathbf{x}$ and the prompts $\mathbf{y}_k$ and pick the indices for top-$r$ scores as the labels of $\mathbf{x}$. The following corollary provides the guarantee of zero-shot transfer learning for CLIP.

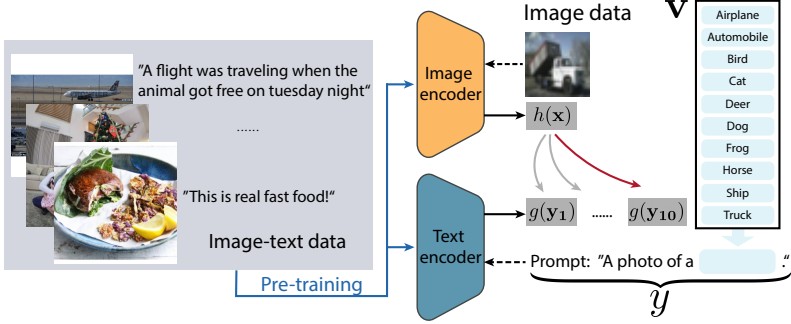

**Figure 2:** Illustration of zero-shot transfer learning. With the encoders jointly pre-trained on the image-text dataset, zero-shot transfer is done by issuing prompts according to all the potential labels of the task. With similarity score computed between the image embedding and all prompt embeddings, the label that resulted in highest similarity is the prediction.

**Corollary 5.1.** Suppose the result of Theorem 4.2 holds for the learned similarity function $\widehat{f}$. Then we calculate the similarity score $\widehat{f}(\mathbf{x}, \mathbf{y}_k)$ for all $k \in [K]$ and pick the indices of the top-$r$ scores within the set $\{\widehat{f}(\mathbf{x}, \mathbf{y}_k)\}$ as the predictions of the image $\mathbf{x}$. Then the top-$r$ error is bounded by $\epsilon'/\log(1+r)$.

In other words, Corollary 5.1 guarantees that a trained CLIP model can achieve small top-$r$ error, where $r$ is an integer usually selected as 1 or 3 in real-data experiments.

**Remark 5.2.** The result in Corollary 5.1 can be generalized to out-of-distribution zero-shot transfer. For example, we can deal with the case where the distribution of the prompts $\mathcal{D}_{\mathbf{y}|\mathbf{v}_k}$ and the image distribution $\mathcal{D}_{\mathbf{x}}$ are shifted. As long as the $\chi^2$ distance between the shifted distributions is bounded, we can provide a top-$r$ error guarantee (see Appendix F for a detailed discussion).

Next, we will introduce a specific problem to illustrate how CLIP can learn transferable features with distinguishable margins, which is hard to achieve by simple square loss.

**Definition 5.3** (A Case Study). Let shared feature $\mathbf{z} \in \mathbb{R}^{K_1}$ be random variable uniformly drawn from the set $\mathcal{V} = \{\mathbf{v}_1, \ldots, \mathbf{v}_K\}$ where $\|\mathbf{v}_k\|_2 = 1$, $\max_{k \neq k'}\langle \mathbf{v}_k, \mathbf{v}'_k \rangle = 1 - \gamma$. Let $\boldsymbol{\xi} \in \mathbb{R}^{K_2}, \boldsymbol{\zeta} \in \mathbb{R}^{K_3}$ be unique random features satisfying $\|\boldsymbol{\xi}\|_2, \|\boldsymbol{\zeta}\|_2 \leq R$ and are mutually independent given $\mathbf{z}$. The image-text pair is generated as

$$\mathbf{x} = \mathbf{G}\begin{bmatrix}\mathbf{z}\\\boldsymbol{\xi}\end{bmatrix} = \mathbf{G}_1\mathbf{z} + \mathbf{G}_2\boldsymbol{\xi}, \qquad \mathbf{y} = \mathbf{H}\begin{bmatrix}\mathbf{z}\\\boldsymbol{\zeta}\end{bmatrix} = \mathbf{H}_1\mathbf{z} + \mathbf{H}_2\boldsymbol{\zeta},$$

where $\mathbf{G} \in \mathbb{R}^{d_1 \times (K_1 + K_2)}$ is the image dictionary with full rank $(K_1 + K_2)$, $\mathbf{H} \in \mathbb{R}^{d_2 \times (K_1 + K_3)}$ is the text dictionary with full rank $(K_1 + K_3)$.

For the distribution in Definition 5.3, locked image-text tuning is enough to learn transferrable features (Zhai et al., 2022). In particular, we choose the score function as $f_{\mathbf{W}} = \langle \mathbf{g}(\mathbf{x}), \mathbf{h}(\mathbf{y}) \rangle$ where the embeddings are $\mathbf{g}(\mathbf{x}) = \mathbf{W}\mathbf{x}, \mathbf{h}(\mathbf{y}) = \mathbf{y}$. Next, we verify Assumptions 4.1 for the specified distribution.

**Lemma 5.4** (Completeness). There exist a score function $f^*(\mathbf{x}, \mathbf{y}) = \langle \mathbf{W}^*\mathbf{x}, \mathbf{y} \rangle$ with $\mathbf{W}^* \in \mathbb{R}^{d_2 \times d_1}$ satisfying
- $|f^*| \leq 1$,
- For $(\mathbf{x}, \mathbf{y}, \mathbf{z}) \sim \mathcal{D}_{\mathbf{x} \times \mathbf{y} \times \mathbf{z}}$, variance $\mathbb{E}_{(\mathbf{y},\mathbf{z})}\big[\mathrm{Var}_{\mathbf{x}|\mathbf{z}}(f^*(\mathbf{x}, \mathbf{y}))\big] = \mathbb{E}_{(\mathbf{x},\mathbf{z})}\big[\mathrm{Var}_{\mathbf{y}|\mathbf{z}}(f^*(\mathbf{x}, \mathbf{y}))\big] = 0$,
- Let $\mathbf{x} \sim \mathcal{D}_{\mathbf{x}|\mathbf{z}}, \mathbf{y} \sim \mathcal{D}_{\mathbf{y}|\mathbf{z}}, \mathbf{x}' \sim \mathcal{D}_{\mathbf{x}'|\mathbf{z}'}, \mathbf{y}' \sim \mathcal{D}_{\mathbf{y}'|\mathbf{z}'}$ where $\mathbf{z} \neq \mathbf{z}'$. With probability 1, we have that $f^*(\mathbf{x}', \mathbf{y}) \leq f^*(\mathbf{x}, \mathbf{y}) - \gamma$ and $f^*(\mathbf{x}, \mathbf{y}') \leq f^*(\mathbf{x}, \mathbf{y}) - \gamma$.

Then we can use the standard gradient descent on the empirical loss to learn the score function $f$, i.e.,

$$\mathbf{W}^{(t+1)} = \mathbf{W}^{(t)} - \eta\nabla_{\mathbf{W}}\widehat{L}_S(f, \tau).$$

The following theorem gives convergence guarantees for CLIP and provides the upper bound of its zero-shot transfer error.

**Theorem 5.5.** For sufficiently large $n$, set the learning rate $\eta = O(\epsilon\tau^2\|\mathbf{G}\|^{-2}\|\mathbf{H}\|_2^{-2}(1 + R)^{-4})$, gradient descent can find $\widehat{\mathbf{W}}$ within $4\|\mathbf{W}^{(0)} - \mathbf{W}^*\|_F^2/(\eta\epsilon)$ iterations such that $L_{\mathcal{D}^B}(\widehat{f}, \tau) \leq L_{\mathcal{D}^B}(f^*, \tau) + \epsilon$ where $\widehat{f} = \langle\widehat{\mathbf{W}}\mathbf{x}, \mathbf{y}\rangle$. In addition, the top-$r$ zero-shot transfer error is bounded by $\epsilon'/\log(1+r)$, where $\epsilon' = (C_B + 2) \cdot \left[\epsilon + 2B\exp(-\gamma/\tau)\right]$ and $C_B = \widetilde{O}(K/B)$.

### 5.1 Square Loss Fails Zero-Shot Learning

Another conceivable method is to use the square loss to align the embeddings of $\mathbf{x}, \mathbf{y}$. Here, we investigate why such simple loss can not successfully learn transferrable representations and reveal the significance of contrastive loss in multi-modal learning. In particular, we use $\mathbb{E}[\|\mathbf{g}(\mathbf{x}) - \mathbf{y}\|_2^2]$ to learn the embedding $\mathbf{g}$. By Lee et al. (2021), we know that the embedding $\mathbf{g}(\mathbf{x})$ indeed preserves the information of the shared feature $\mathbf{z}$ and can be used to predict the label $k$ (the index of $\mathbf{z}$ in the dictionary) using linear probing with additional $\widetilde{O}(K)$ examples $\{(k, \mathbf{x}), \mathbf{x} \sim \mathcal{D}_{\mathbf{x}|\mathbf{v}_k}\}$. Given the success of $\mathbf{g}$ as a representation for the downstream classification problem, a natural question arises: Can the learned embedding be used for the *zero-shot transfer* task, using only $K$ prompts $\mathbf{y}_k, k \in [K]$ where $\mathbf{y}_k \sim \mathcal{D}_{\mathbf{y}|\mathbf{v}_k}$?

Surprisingly, the answer is negative. We find that even if we can train with population risk and get the Bayesian optimal predictor, the learned representation $\mathbf{g}$ is not suitable for the zero-shot transfer.

To make a fair comparison, we also consider the data distribution introduced in Definition 5.3 and present the following results.

**Theorem 5.6.** The Bayesian optimal representation $\mathbf{g}$ is $\mathbf{g}(\mathbf{x}) = \mathbf{H} \begin{bmatrix} \mathbf{z} \\ \mathbb{E}[\boldsymbol{\zeta}|\mathbf{z}] \end{bmatrix}$.

Since $\mathbb{E}[\boldsymbol{\zeta}|\mathbf{z}]$ lies in the unique feature space, the accuracy of zero-shot learning can be largely determined by the unique features $\boldsymbol{\zeta}$, i.e., the quality of the prompt. In detail, given a set of prompts $\{\mathbf{y}_k\}$, we evaluate the similarity between representations $\mathbf{g}(\mathbf{x})$ and $\mathbf{h}(\mathbf{y}_k) = \mathbf{y}_k$ under different similarity scores, including (1) inner product similarity: $f(\mathbf{x}, \mathbf{y}_k) = \langle \mathbf{g}(\mathbf{x}), \mathbf{y}_k \rangle$; (2) cosine similarity: $f(\mathbf{x}, \mathbf{y}_k) = \langle \mathbf{g}(\mathbf{x})/\|\mathbf{g}(\mathbf{x})\|_2, \mathbf{y}_k/\|\mathbf{y}_k\|_2 \rangle$; and (3) $L_2$ similarity: $(-1) \cdot \|\mathbf{g}(\mathbf{x}) - \mathbf{h}(\mathbf{y}_k)\|_2$. The following corollary formally states the negative result.

**Corollary 5.7.** For the distribution in Definition 5.3 with $\mathbf{H} = \begin{bmatrix} \mathbf{I} \\ \mathbf{0} \end{bmatrix}$, margin $\gamma < 1/3$, text unique feature $\boldsymbol{\zeta} \in \mathbb{R}^{K_3}$ drawn from $\{\mathbf{e}_1, \mathbf{e}_2\}$ with probability $1/3, 2/3$ respectively. Then, the zero-shot top-1 error is at least $1/(3K)$ regardless of the three similarity scores.

**Remark 5.8.** By Theorem 5.5, we can achieve arbitrarily small top-1 error by CLIP as long as $\epsilon$ and $\tau$ are sufficiently small. However, for the representation learned from the square loss, the top-1 error is at least a constant even if we can achieve the Beyasian optimal predictor.

## 6 LEARN BETTER REPRESENTATION VIA REGULARIZATION

In Corollary 5.1, we know that CLIP can achieve a small error for zero-shot transfer tasks. In this section, we investigate how large the margin can be achieved between different features $\mathbf{z}$'s. Under the same condition of Corollary 5.1, we present the following corollary.

**Corollary 6.1.** Suppose the result of Theorem 4.2 holds for the learned similarity function $\widehat{f}$. We calculate the similarity score $\widehat{f}(\mathbf{x}, \mathbf{y}_k)$ for all $k \in [K]$. Then with probability at least $1 - 4\epsilon'$, the top-1 result gives the correct answer with a margin $\tau$.

Here, the margin depends on the temperature parameter $\tau$. Note that we only achieve the margin with $\tau$ instead of $\gamma$ guaranteed in the Assumption 4.1. Therefore, CLIP needs to choose $\tau \ll \gamma$ to ensure a good performance, indicating a theoretical gap for the learned margin. To further investigate this gap, we consider the simple case study in Definition 5.3 and have the following negative result.

**Theorem 6.2.** Under the same condition as Theorem 5.5, there exists a special case with initialization $\mathbf{W}^{(0)}$, such that when we train the model with polynomial iterations $T = \text{poly}(\eta^{-1}, \epsilon, d_1, d_2)$, with probability at least $0.99$, the top-1 result can only give the correct answer with a margin $\widetilde{O}(\tau)$.

Such a phenomenon also exists in real data: the margin will decrease when temperature $\tau$ decreases (see Figure 3). The reason is that softmax function $L(\mathbf{a}) = \log(\sum_i \exp(a_i))$ is convex but not strongly convex and has an exponential-decaying tail. Once the score function $f$ with the features $\mathbf{g}$ and $\mathbf{h}$ achieves the margin of order $\Omega(\tau)$, the gradient will exponentially decrease. Therefore, the weight will not be updated effectively. To obtain a larger margin, it is natural to add the following regularization to maximize the score of the positive pairs and minimize the score of the negative pairs.

$$R(f) = \frac{1}{|S^-|} \sum_{(\mathbf{x}, \mathbf{y}') \in S^-} f(\mathbf{x}, \mathbf{y}') - \frac{1}{|S^+|} \sum_{(\mathbf{x}, \mathbf{y}') \in S^+} f(\mathbf{x}, \mathbf{y}'), \tag{6.1}$$

where $S_+$ is the set of positive pairs that have the same shared feature $\mathbf{z} = \mathbf{z}'$, and $S_-$ is the set of the negative pairs that have different shared feature $\mathbf{z} \neq \mathbf{z}$. However, the set $S_-$ is very hard to determine since different image-text pairs in the batch can possibly have the same shared features, as we demonstrated in Figure 1. On the other hand, the set of $S_+$ can be simply chosen as the training data set $S$. Therefore, we propose to use only one direction in (6.1) as the regularization, i.e.,

$$R(f) = -\frac{1}{|S|} \sum_{(\mathbf{x}, \mathbf{y}) \in S} f(\mathbf{x}, \mathbf{y}).$$

In particular, when $\mathbf{g}$ and $\mathbf{h}$ are normalized representations with unit $L_2$ norm and we use inner product similarity $f(\mathbf{x}, \mathbf{y}) = \langle \mathbf{g}(\mathbf{x}), \mathbf{h}(\mathbf{y}) \rangle$, our regularization can be viewed as the $L_2$ distance between the embeddings since

$$R_S(f) = \frac{1}{2|S|} \sum_{(\mathbf{x}, \mathbf{y}) \in S} \left[ \|\mathbf{g}(\mathbf{x})\|_2^2 + \|\mathbf{h}(\mathbf{y})\|_2^2 - 2\langle \mathbf{g}(\mathbf{x}), \mathbf{h}(\mathbf{y}) \rangle \right] - 1$$

$$= \frac{1}{2|S|} \sum_{(\mathbf{x},\mathbf{y}) \in S} \|\mathbf{g}(\mathbf{x}) - \mathbf{h}(\mathbf{y})\|_2^2 - 1.$$

Similarly, for a sampled batch $S'$, the regularized loss is defined as $\widehat{L}_{S'}(f, \tau, \lambda) = L_{S'}(f, \tau) + \lambda \cdot R_{S'}(f)$, where $\lambda > 0$ is a small regularization parameter. The following theorem shows that the regularization will provably improve the margin.

**Theorem 6.3.** Under the same condition as Theorem 6.2, with sufficiently small $\tau$ and appropriately chosen $\lambda$, within polynomial iterations $T = \text{poly}(\eta^{-1}, \epsilon, d_1, d_2)$, we can find a score function $\widehat{f}$ with large margin. In particular, with a probability of at least 0.99, the top-1 result gives the correct label with a margin $\widetilde{\Omega}(\gamma)$.

Recall in Theorem 6.2, where the vanilla model achieves margin of $\widetilde{O}(\tau)$, the regularization term provably improves the margin to $\widetilde{\Omega}(\gamma)$. Lastly, our regularization term shares similar concept as SimSiam (Chen & He, 2021), which only considers the positive pairs in the single modality setting.

# 7 EXPERIMENTS

In this section, we present experiment results on real datasets to verify our theoretical findings. Accordingly, we examine our new CLIP-like training objective and showcase its improvement in performance on diverse zero-shot transfer and linear probing tasks.

**Datasets.** For performance evaluation, we primarily focus on Conceptual Captions 3M (CC3M) (Sharma et al., 2018) as the pretraining dataset, in alignment with prior literature (Li et al., 2022; Goel et al., 2022). Additionally, we use MSCOCO (Chen et al., 2015) in order to conduct lightweight real data experiments to validate our theoretical findings.

**Architectures.** We consider the same setting for experiments on all baseline CLIP-objectives. Following the original CLIP paper, we employ ResNet (He et al., 2016) as the image encoder and the Transformer architecture (Vaswani et al., 2017) as the text encoder. We utilize pre-trained weights for both encoders to achieve faster convergence. These include the pre-trained ResNet-50 from the PyTorch Image Models library (Wightman, 2019) and pre-trained DistilBERT from the Huggingface Transformers library (Wolf et al., 2020). We note that, the setting of training from pre-trained weights is also considered in several previous literature (Zhai et al., 2022; Alayrac et al., 2022). Lastly, our experiments can be feasibly ran on a single GeForce RTX 2080 GPU. Detailed hyperparameters and additional experiments are presented in Appendix C.

## 7.1 EFFECT OF TEMPERATURE ON MARGIN

In support of our theoretical discussions in Corollary 6.1 and Theorem 6.2 that find the positive correlation between the margin and the temperature parameter, we conduct real data experiments to confirm the impact of temperature on the margin. In Figure 3, we examine the margin distribution of CLIP models trained at varying temperatures. Specifically, the margin is evaluated by the difference between a diagonal value and an off-diagonal value within a batch: $f(\mathbf{x}_i, \mathbf{y}_i) - f(\mathbf{x}_j, \mathbf{y}_i)$ and $f(\mathbf{x}_i, \mathbf{y}_i) - f(\mathbf{x}_i, \mathbf{y}_j)$ (see Appendix A for details). We collect the results of untrained and trained CLIP models on all batches within the MSCOCO training dataset with batch size 64.

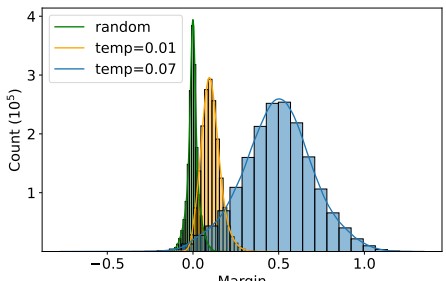

**Figure 3:** The distribution of the margins with regard to CLIP models trained at different temperature values. Margin is computed within each batch of the data.

As depicted in Figure 3, a CLIP model with random initialization at the projection layers has margins normally distributed near zero, whereas trained models exhibit positive margins, signifying successful training. Furthermore, we consider CLIP models trained at fixed temperature values of 0.07 (the default starting value for the original CLIP) and 0.01 (the clipping value). As observed in the figure, the margin distribution shifts to the left as temperature $\tau$ decreases, suggesting that a extremely small $\tau$ leads to small margins, aligning with the results in Corollary 6.1.

## 7.2 ZERO-SHOT TRANSFER

To confirm Theorem 6.3, we investigate the advantages of incorporating our regularization term during training by evaluating zero-shot transfer accuracy and linear probing on various datasets. We consider the following training objectives when adding our regularization: (1) the original CLIP (Radford et al., 2021), and (2) CyCLIP (Goel et al., 2022) with cross-modal and in-modal consistency regularizations,

adopting the same hyperparameters for the regularizations as outlined in Goel et al. (2022). All models are trained on CC3M using the same model architecture, batch size, and optimizer settings. Further experimental details are provided in Appendix C.

In Table 1, we present the zero-shot test accuracy of CLIP models trained with the original CLIP objective and the CyCLIP objective. Firstly, we demonstrate the model's performance when training solely on the regularization term (L2) and compare to that of the CLIP objective. In alignment with our Corollary 5.7, we can observe on real data that training exclusively on the L2 objective leads to a large error and even random guessing on the zero-shot datasets. Combining with our theoretical analysis, we show that a naive square loss fails to learn transferable representations. In the context of multi-modal learning, contrastive loss is important. Moreover, confirming our result from Theorem 6.3, incorporating the regularization term into the contrastive objective effectively enhances performance across the majority of zero-shot transfer tasks. It improves over the baseline on 5 out of 6 datasets by a good margin. The best performance achieved by adding regularization to the CLIP objective outperforms its original objective by 3.62% on CIFAR10 and by 2.06% on average of all datasets.

In Table 2, we report the results of linear probing, where logistic regression classifiers are fitted to the embeddings learned by the image encoders of our compared models. This table offers an assessment of the visual representation learning for each training objective. Similarly supporting Corollary 5.7, training on the regularization term only results in learning bad representations that yield unsatisfactory performances on linear probing. Moreover, in alignment with Theorem 6.3, we observe that adding the regularization term consistently improves CLIP's performance across various datasets by an average of 1.54%.

**Table 1:** Zero-shot top-1 accuracy (%). Notably, adding the regularization term successfully improves the baselines on 5 out of the 6 datasets.

|  | CIFAR10 | CIFAR100 | STL10 | Food101 | ImageNetV2 | DTD | Average |
|---|---|---|---|---|---|---|---|
| Reg | 10.04 | 1.05 | 9.95 | 1.08 | 0.11 | 2.07 | 3.47 |
| CLIP | 63.85 | 31.17 | 90.35 | 8.39 | 20.24 | **21.22** | 39.20 |
| CyCLIP | 60.71 | 28.87 | 89.98 | 9.72 | 19.66 | 20.21 | 38.19 |
| CLIP+Reg | **67.47** | **33.33** | **92.64** | **12.14** | **22.36** | 19.63 | **41.26** |

**Table 2:** Linear probing accuracy (%). All logistic regression models are trained till convergence. Adding our regularization term to CLIP provides decent improvements across all datasets. On CyCLIP, we also makes improvements on the majority of datasets.

|  | CIFAR10 | CIFAR100 | STL10 | Food101 | DTD | Flowers | OxfordPets | Average |
|---|---|---|---|---|---|---|---|---|
| Reg | 14.09 | 2.17 | 17.86 | 1.73 | 3.40 | 2.18 | 4.12 | 6.51 |
| CLIP | 87.30 | 66.03 | 93.26 | 62.8 | 56.70 | 70.24 | 72.91 | 72.75 |
| CyCLIP | 86.31 | 63.93 | 93.69 | 61.57 | 56.86 | 70.56 | 70.46 | 71.91 |
| CLIP+Reg | **88.49** | **66.16** | **94.98** | **63.39** | **57.66** | **72.21** | **77.13** | **74.29** |

## 8 CONCLUSION

In this paper, we rigorously investigated the theoretical underpinnings of transferable representation learning in CLIP, addressing the challenges associated with feature domain alignment and shared feature sparsity. We provided insights through detailed examination of specific cases and corroborated our theory with empirical evidence. Lastly, we proposed a regularization term grounded in our theoretical findings to enhance CLIP's performance in various downstream tasks, including zero-shot transfer and linear probing. Combining rigorous theoretical analysis with empirical validation, we contribute to the advancement of understanding in multi-modal contrastive learning.

**Limitations and future work.** We emphasize that our primary contribution lies in providing theoretical insights into transferable representation learning in CLIP, which assumes a one-to-one mapping between image-text pairs. Interesting future works include extending the analysis to more modalities and exploring other multimodal training algorithms. Another limitation of our work is the limited computational resources, where we used relatively smaller training data than the large-scale web data used by CLIP and are also restricted to smaller training batch sizes as compared to industry standards.

ACKNOWLEDGEMENT

We thank the anonymous reviewers and area chair for their helpful comments. ZC, YD and QG are supported in part by the National Science Foundation CAREER Award 1906169, IIS-2008981, CHE-2247426 and the Sloan Research Fellowship. The views and conclusions contained in this paper are those of the authors and should not be interpreted as representing any funding agencies.

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

# A DISCUSSION ON THE MARGIN IN CLIP

"Margin" plays an important role in unimodal contrastive learning (Wang & Liu, 2021), which measures the desired similarity difference between positive and negative pairs in the learned feature space: $f(\mathbf{x}, \mathbf{x}^+) - f(\mathbf{x}, \mathbf{x}^-)$. This metric ensures that the similarity of positive pair representations exceeds a specific threshold, while preserving a greater distance for the negative pairs. In practice, a large margin encourages the model to learn meaningful and discriminative data representations, thereby achieving better results in the downstream task (Chen et al., 2021).

In exploring the CLIP model, we focus on the concept of margin from a multi-modal perspective. For two independent tuple $(\mathbf{x}, \mathbf{y}, \mathbf{z}) \sim \mathcal{D}_{\mathbf{x} \times \mathbf{y} \times \mathbf{z}}$ and $(\mathbf{x}', \mathbf{y}', \mathbf{z}') \sim \mathcal{D}_{\mathbf{x} \times \mathbf{y} \times \mathbf{z}}$, we formally introduce a measure as follows

$$\alpha_\gamma = \mathbb{P}\Big(\mathbf{z} \neq \mathbf{z}', f(\mathbf{x}, \mathbf{y}) - f(\mathbf{x}, \mathbf{y}') \leq \gamma\Big) + \mathbb{P}\Big(\mathbf{z} \neq \mathbf{z}', f(\mathbf{x}, \mathbf{y}) - f(\mathbf{x}', \mathbf{y}) \leq \gamma\Big) \tag{A.1}$$

where $\gamma$ denotes the margin, and $\alpha_\gamma$ is failure probability of failing to achieve this margin. We note that when $\mathbf{z} = \mathbf{z}'$, $\mathbf{x}, \mathbf{y}, \mathbf{x}', \mathbf{y}'$ will form positive pairs, thus excluded in equation (A.1). Unfortunately, we can access $\mathcal{D}_{\mathbf{x} \times \mathbf{y}}$ in real applications but have limited knowledge of the latent variable $\mathbf{z}$. This limitation complicates the identification of all positive pairs within a batch of data.

## A.1 MARGIN AND VISUAL-SEMANTIC ALIGNMENT

When $\mathbf{g}$ and $\mathbf{h}$ are normalized representations with unit $L_2$ norm and we use inner product similarity $f(\mathbf{x}) = \langle \mathbf{g}(\mathbf{x}), \mathbf{h}(\mathbf{y}) \rangle$. The formula $f(\mathbf{x}, \mathbf{y}) - f(\mathbf{x}, \mathbf{y}')$ can be expressed as

$$f(\mathbf{x}, \mathbf{y}) - f(\mathbf{x}, \mathbf{y}') = \frac{1}{2}\big[2 - \|\mathbf{g}(\mathbf{x}) - \mathbf{h}(\mathbf{y})\|_2^2\big] - \frac{1}{2}\big[2 - \|\mathbf{g}(\mathbf{x}) - \mathbf{h}(\mathbf{y}')\|_2^2\big]$$
$$= \frac{1}{2}\big[\ \underbrace{\|\mathbf{g}(\mathbf{x}) - \mathbf{h}(\mathbf{y}')\|_2^2}_{\text{Negative}-\text{pair Distance}}\ -\ \underbrace{\|\mathbf{g}(\mathbf{x}) - \mathbf{h}(\mathbf{y})\|_2^2}_{\text{Positive}-\text{pair Distance}}\ \big], \tag{A.2}$$

where the second equality uses the property of unit $L_2$ norm. By (A.2), we can see that a larger margin value implies that the embeddings $\mathbf{g}$ and $\mathbf{h}$ of the positive pairs remain in closer proximity, while the embeddings of negative pairs are far away from each other. This is a crucial aspect of contrastive learning, especially when considering the CLIP model.

In unimodal contrastive learning, $\mathbf{y} = \mathbf{x}^+$ typically follows the same distribution of $\mathbf{x}$, and $\mathbf{h}$ is chosen to be identical to $\mathbf{g}$. Consequently, the embedding difference $\mathbf{g}(\mathbf{x}) - \mathbf{h}(\mathbf{y})$ will generally exhibit a zero mean. In this scenario, the variance of the embedding, rather than its mean, becomes the dominant term for positive-pair distance in (A.2). However, this is not the case for the CLIP model since $\mathbf{x}, \mathbf{y}$ belong to different modalities, and thus $\mathbf{h}$ is no longer chosen to be identical to $\mathbf{g}$.

Moreover, identifying negative pairs in a batch for image-text data is challenging. To **empirically** mitigate the issue, Yang et al. (2022) proposed UniCL for multi-modal contrastive learning. Unlike vanilla CLIP, UniCL additionally consider image-label data and group these data with identical classes, which facilitates negative pair identification within the dataset. However, this strategy necessitates additional group information about the dataset, being either class label or concept. Our paper aims to **theoretically** tackle the identification problem by integrating this grouping mismatch into our analysis. We recognize the significance of empirically addressing this issue like Yang et al. (2022), but it goes beyond the scope of current work.

A larger margin of $f$ indicates an improved visual-semantic alignment. Thus, we favor a function $f$ that achieves a larger margin $\gamma$ with a smaller $\alpha_\gamma$. Under Assumption 4.1, we define the $(\alpha, \beta, \gamma)$ completeness, ensuring the existence of such a function. To find a function with a larger margin more effectively, we introduce a new regularizer in Section 6, specifically tailored for the CLIP model. This regularization approach does not require identifying negative pairs and is particularly suitable for CLIP, as it only penalizes the positive-pair distance $\|\mathbf{g}(\mathbf{x}) - \mathbf{h}(\mathbf{y})\|_2^2$

Chen et al. (2021) proposes a novel large-margin contrastive learning (LMCL) method in unimodal contrastive learning, regularizing both positive and negative pair distances. In our study, we choose to regularize only the positive pair distance, acknowledging the unique characteristics of the CLIP model: different embedding functions $\mathbf{g}, \mathbf{h}$ for images and texts and the difficulty in identifying negative pairs. We also conducted an ablation study for only regularizing the off-diagonal term in the batch. We find that off-diagonal pair regularization yields marginal improvements in downstream zero-shot tasks and lacks stability compared to the regularizer proposed in Section 6 (detailed in Section C.2).

A.2   ESTIMATION OF THE MARGIN

In this subsection, we will discuss how to verify the Assumption 4.1 and measure the quality of the learned function with margin. We introduce an approximate measure $\widehat{\alpha}_\gamma$ as follows,

$$\widehat{\alpha}_\gamma = \mathbb{P}\Big(f(\mathbf{x}, \mathbf{y}) - f(\mathbf{x}, \mathbf{y}') \leq \gamma\Big) + \mathbb{P}\Big(f(\mathbf{x}, \mathbf{y}) - f(\mathbf{x}', \mathbf{y}) \leq \gamma\Big) \tag{A.3}$$

$\widehat{\alpha}_\gamma$ differs from the $\alpha_\gamma$ since we didn't extinguish different classes in the probability. Therefore we can easily calculate $\widehat{\alpha}_\gamma$ without observe $\mathbf{z}$. In practice, (A.3) can be evaluated by the difference between a diagonal value and an off-diagonal value within a batch: $f(\mathbf{x}_i, \mathbf{y}_i) - f(\mathbf{x}_j, \mathbf{y}_i)$ and $f(\mathbf{x}_i, \mathbf{y}_i) - f(\mathbf{x}_i, \mathbf{y}_j)$ (as illustrated in Figure 3).

Moreover, we have the following upper and low bounds, which show that $\widehat{\alpha}_\gamma$ can approximate $\alpha_\gamma$.

**Theorem A.1.** Let $\gamma \geq 0$, then we have that

$$\widehat{\alpha}_\gamma \geq \alpha_\gamma \geq \widehat{\alpha}_\gamma - \sum_{k \in [K]} p_k^2.$$

where $p_k$ is the probability of the classes in Assumption 3.1. Besides, the second inequality becomes exact equality for $\gamma = 0$.

*Proof.*

$$\widehat{\alpha}_\gamma = \mathbb{P}\Big(f(\mathbf{x}, \mathbf{y}) - f(\mathbf{x}, \mathbf{y}') \leq \gamma\Big) + \mathbb{P}\Big(f(\mathbf{x}, \mathbf{y}) - f(\mathbf{x}', \mathbf{y}) \leq \gamma\Big)$$

$$= \underbrace{\mathbb{P}\Big(\mathbf{z} \neq \mathbf{z}', f(\mathbf{x}, \mathbf{y}) - f(\mathbf{x}, \mathbf{y}') \leq \gamma\Big) + \mathbb{P}\Big(\mathbf{z} \neq \mathbf{z}', f(\mathbf{x}, \mathbf{y}) - f(\mathbf{x}', \mathbf{y}) \leq \gamma\Big)}_{=\alpha_\gamma}$$

$$+ \underbrace{\mathbb{P}\Big(\mathbf{z} = \mathbf{z}', f(\mathbf{x}, \mathbf{y}) - f(\mathbf{x}, \mathbf{y}') \leq \gamma\Big) + \mathbb{P}\Big(\mathbf{z} = \mathbf{z}', f(\mathbf{x}, \mathbf{y}) - f(\mathbf{x}', \mathbf{y}) \leq \gamma\Big)}_{\text{Approximate Error}}.$$

The Approximate Error has a naive lower bound of $0$, and we can upper bound it as follows

$$\mathbb{P}\Big(\mathbf{z} = \mathbf{z}', f(\mathbf{x}, \mathbf{y}) - f(\mathbf{x}, \mathbf{y}') \leq \gamma\Big) = \mathbb{P}\Big(f(\mathbf{x}, \mathbf{y}) - f(\mathbf{x}, \mathbf{y}') \leq \gamma | \mathbf{z} = \mathbf{z}'\Big) \cdot \mathbb{P}(\mathbf{z} = \mathbf{z}')$$

$$\leq \mathbb{P}\Big(f(\mathbf{x}, \mathbf{y}) - f(\mathbf{x}, \mathbf{y}') \leq 0 | \mathbf{z} = \mathbf{z}'\Big) \cdot \mathbb{P}(\mathbf{z} = \mathbf{z}')$$

$$= 1/2 \sum_{k \in [K]} p_k^2.$$

were the the inequality is due to fact that $\gamma \geq 0$ and the last equality is because $\mathbf{y}'$ and $\mathbf{y}$ are symmetric give $\mathbf{z} = \mathbf{z}'$. Finally, the inequality is an exact equality for $\gamma = 0$.  □

By Theorem A.1, $\alpha_\gamma$ and $\widehat{\alpha}_\gamma$ are close to each other if $\max_{k \in [K]} p_k$ is small, since

$$\sum_{k \in [K]} p_k^2 \leq \sum_{k \in [K]} p_k \cdot \max_{k \in [K]} p_k = \max_{k \in [K]} p_k \cdot \Big(\sum_{k \in [K]} p_k\Big) = \max_{k \in [K]} p_k.$$

**Relation with the Figure 6:** $\widehat{\alpha}_\gamma$ has a strong relationship with Figure 6, where we have plot the distribution of $f(\mathbf{x}, \mathbf{y}) - f(\mathbf{x}, \mathbf{y}')$ and $f(\mathbf{x}, \mathbf{y}) - f(\mathbf{x}', \mathbf{y})$. The figure can be viewed as the figure of the probability density function, and $\widehat{\alpha}_\gamma$ can be viewed as the cumulative probability function, which is the integral of probability mass smaller than $\gamma$. From Figure 6, we can deduce that the CLIP learned with regularization has consistently smaller $\widehat{\alpha}_\gamma$ for all $\gamma \geq 0$.

B   DISCUSSION ON THE TRAINABLE TEMPERATURE PARAMETER $\tau$

This section considers the setting where the temperature $\tau$ is also trainable with the following loss.

$$L_{\mathcal{D}^B}(f, \tau) = \mathbb{E}\Bigg[\log\Big(\sum_{t \in [B]} \exp\big([f(\mathbf{x}_1, \mathbf{y}_t) - f(\mathbf{x}_1, \mathbf{y}_1)]/\tau\big)\Big)\Bigg]$$

$$+ \mathbb{E}\Bigg[\log\Big(\sum_{t \in [B]} \exp\big([f(\mathbf{x}_t, \mathbf{y}_1) - f(\mathbf{x}_1, \mathbf{y}_1)]/\tau\big)\Big)\Bigg].$$

Suppose $\tau$ is clipped to be within the range $[\tau_{\min}, \tau_{\max}]$, it is natural to assume that we can obtain function $\widehat{f}$ with temperature $\widehat{\tau} \in [\tau_{\min}, \tau_{\max}]$ such that

$$L_{\mathcal{D}^B}(\widehat{f}, \widehat{\tau}) \leq \min_{\tau \in [\tau_{\min}, \tau_{\max}]} L_{\mathcal{D}^B}(f^*, \tau) + \epsilon \tag{B.1}$$

$$= L_{\mathcal{D}^B}(f^*, \widehat{\tau}) + \epsilon - \left( L_{\mathcal{D}^B}(f^*, \widehat{\tau}) - \min_{\tau \in [\tau_{\min}, \tau_{\max}]} L_{\mathcal{D}^B}(f^*, \tau) \right) \tag{B.2}$$

$$= L_{\mathcal{D}^B}(f^*, \widehat{\tau}) + \widetilde{\epsilon} \tag{B.3}$$

where $\widetilde{\epsilon} = \epsilon - \left( L_{\mathcal{D}^B}(f^*, \widehat{\tau}) - \min_{\tau \in [\tau_{\min}, \tau_{\max}]} L_{\mathcal{D}^B}(f^*, \tau) \right) \leq \epsilon$. Since $\widetilde{\epsilon}$ is smaller than $\epsilon$, we can get smaller $\epsilon'$ in Theorem 4.2, and thus get smaller top-r error in zero-shot transfer task by Corollary 5.1. This observation implies that the representation $(\widehat{f}, \widehat{\tau})$ found by trainable temperature can be better than the representation $(\widehat{f'}, \widehat{\tau})$ found with fixed temperature $\widehat{\tau}$.

## C  ADDITIONAL EXPERIMENT RESULTS

We consider the same model architecture as CLIP (Radford et al., 2021) and consider ResNet-50 (He et al., 2016) as the image encoder and transformer (Vaswani et al., 2017) architecture as the text encoder. Specifically, we use pre-trained weights for the encoders for faster convergence in training. We follow the code framework in Shariatnia (2021) and use pre-trained ResNet-50 from the PyTorch Image Models library (Wightman, 2019) and pre-trained DistilBERT from the Huggingface Transformers library (Wolf et al., 2020). We further have linear projection layers on both image and text encoders, the same as in CLIP, and consider the embedding dimension to be $512$. As we are training at small-scale data with pre-trained encoders, we follow Shariatnia (2021) and use AdamW optimizer with learning rate 1e-4 on the image encoder, 1e-5 on the text encoder, and 1e-3 on the projection layers, with weight decay coefficient 1e-3. Our code is provided anonymously on Github[*].

### C.1  IMAGE-TEXT RETRIEVAL

We additionally consider the image-to-text and text-to-image retrieval downstream tasks in the zero-shot setting. Following the setting outlined by Goel et al. (2022), we use Flickr30K (Plummer et al., 2015) and MSCOCO (Chen et al., 2015) datasets, which are well-established benchmarks for image-text retrieval tasks. We similarly focus on the test data from the Karpathy (Karpathy & Fei-Fei, 2015) split, with Flickr30K comprising 1k test instances and MSCOCO containing 5k. Consistent with the findings of Goel et al. (2022), we observe that text retrieval for a given image tends to be less challenging than image retrieval for a given caption. This is due to the nature of both datasets, where each image is associated with $5$ captions. Our results, as detailed in Table 3 and Table 4, align with this trend. Notably, while CyCLIP does not consistently outperform CLIP, adding our regularization term consistently enhances the performance of both the CLIP and CyCLIP.

**Table 3:** Zero-shot image-to-text and text-to-image retrieval results on Flickr30K test set for CLIP with different regularization techniques (CyCLIP, our regularization, or both).

|  | Text R@1 | Text R@5 | Text R@10 | Image R@1 | Image R@5 | Image R@10 | Average |
|---|---|---|---|---|---|---|---|
| CLIP | 87.36 | 93.0 | 95.18 | 26.88 | 54.18 | 66.22 | 70.47 |
| CLIP+Reg | **87.42** | **93.42** | **95.82** | **29.94** | **58.00** | **69.82** | **72.40** |
| CyCLIP | 87.34 | 93.12 | 95.04 | 29.00 | 56.50 | 67.62 | 71.44 |
| CyCLIP+Reg | 87.20 | 93.20 | 95.56 | 29.14 | 56.94 | 68.64 | 71.78 |

**Table 4:** Zero-shot image-to-text and text-to-image retrieval results on MSCOCO test set for CLIP with different regularization techniques (CyCLIP, our regularization, or both).

|  | Text R@1 | Text R@5 | Text R@10 | Image R@1 | Image R@5 | Image R@10 | Average |
|---|---|---|---|---|---|---|---|
| CLIP | 81.19 | 83.21 | 84.42 | 4.73 | 11.66 | 15.93 | 46.86 |
| CLIP+Reg | **81.25** | **83.31** | 84.49 | 4.98 | 12.14 | 16.66 | 47.14 |
| CyCLIP | 81.06 | 82.92 | 84.28 | 4.70 | 11.66 | 15.93 | 46.86 |
| CyCLIP+Reg | 81.31 | 83.28 | **84.65** | **5.27** | **12.17** | **16.70** | **47.23** |

---

[*]https://anonymous.4open.science/r/CLIP_theory-BC8F/README.md

## C.2 DISCUSSION ON THE "NEGATIVE" PAIRS

As previously discussed in Figure 1 and Section 6, the use of unlabeled image-text data in CLIP pre-training may lead to batches containing off-diagonal pairs that are not genuinely negative. In contrast, in the unimodal setting (Chen et al., 2021), accurately identifying truly negative pairs is more straightforward due to the availability of class labels. However, treating all off-diagonal pairs as negatives in the CLIP framework may not be ideal. We investigate taking off-diagonal pairs within a batch as "negative" pairs and sum them into a regularization term. Again, during the training, we consider sample a batch of image-captions pairs $S' = \{\mathbf{x}_i, \mathbf{y}_i\}_{i=1}^{B} \subseteq S$. The regularization term for the negative pairs is thus

$$R(f) = \lambda \cdot \sum_{i \in S'} \sum_{j \in S', j \neq i} f(\mathbf{x}_i, \mathbf{y}_j),$$

where $\lambda > 0$ is the regularization parameter. In experiments, we let $\lambda = 0.1/(B^2 - B)$ and all the other settings remain the same as our previous experiments. In Table 5, our results show that while positive pair regularization markedly improves performance, off-diagonal pair regularization yields only marginal enhancements on some datasets and no improvement on others. This unstable performance may be attributed to the presence of positive pairs among the off-diagonal elements in the unlabeled image-text data.

**Table 5:** Zero-shot top-1 accuracy (%) with regularization on positive image-text pairs and "negative" pairs.

|          | CIFAR10 | CIFAR100 | STL10 | Food101 | ImageNetV2 | DTD   | Average |
|----------|---------|----------|-------|---------|------------|-------|---------|
| CLIP     | 63.85   | 31.17    | 90.35 | 8.39    | 20.24      | **21.22** | 39.20  |
| CLIP+Pos | **67.47** | **33.33** | **92.64** | **12.14** | **22.36** | 19.63 | **41.26** |
| CLIP+Neg | 64.36   | 31.01    | 91.25 | 9.59    | 20.17      | 20.74 | 39.52   |

## C.3 INVESTIGATION INTO THE IMAGE-CAPTION DATA

In Figure 4, we focus on the MSCOCO image-caption dataset, specifically examining the existence of objects present in images but omitted in their corresponding captions. We found that a significant portion of the data pairs contain at least one such object missing from the caption. In Figure 5,

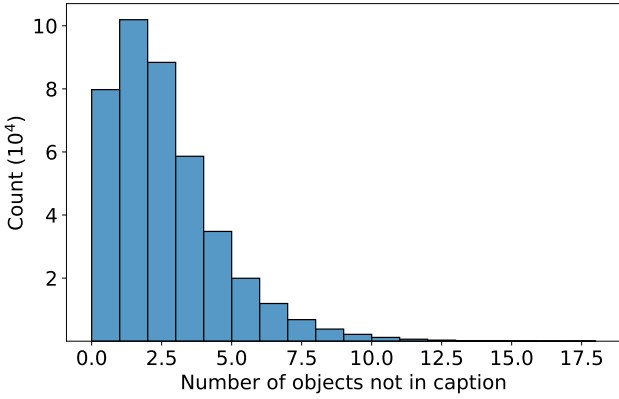

**Figure 4:** Distribution of the image-caption pairs in MSCOCO, where we count the number of object that appeared in the image but was absent from the captions.

we present a random selection of the image-caption pairs in CC3M dataset. These examples are illustrative of the whole dataset, although we cannot provide an exhaustive representation of the numerous examples within the dataset.

## C.4 EFFECT OF TEMPERATURE ON MARGIN

**Setup.** For lightweight exploration in section 7.1, we use the training dataset from MSCOCO (Chen et al., 2015) Image Captioning Task as the data for vision-language contrastive pre-training. Specifically, the dataset contains $82,783$ images where each image is coupled with $5$ captions. We consider each image-caption pair as a data example in pre-training and therefore arrive at $413,915$ pre-training data pairs. We further randomly split the data to keep $20\%$ of the data as validation set and stops

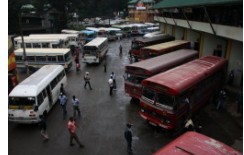

"A very typical bus station."
Missing: people, building, ect.

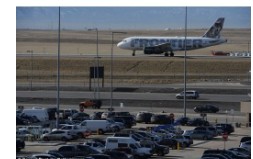

"A flight was traveling when the animal got free on tuesday night."
Missing: cars, road, ect.

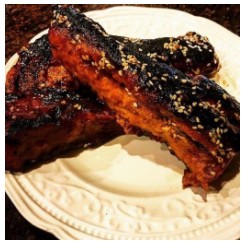

"Venture funded company called these sugar-glazed , slow-smoked ribs."
Missing: plate, table, ect.

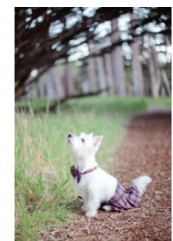

"Dog in a kilt :)"
Missing: grass, tree, ect.

**Figure 5:** Examples of the image-text pairs from CC3M. We identify a few missing visual objects in the captions.

training as the contrastive loss on validation data no longer decreases to avoid overfitting on the small dataset.

**Margin.** Given a training data batch, the margin is consider as the difference between a diagonal value and an off-diagonal value: $f(\mathbf{x}_i, \mathbf{y}_i) - f(\mathbf{x}_j, \mathbf{y}_i)$ and $f(\mathbf{x}_i, \mathbf{y}_i) - f(\mathbf{x}_i, \mathbf{y}_j)$. We consider CLIP models trained at fixed temperature $\tau = 0.07$ and $\tau = 0.01$. We note that $0.07$ is the default value for $\tau$ to start training in CLIP and $0.01$ is the clamping value (equivalently as the maximum logit scale of $4.6052$.) In Figure 3, we collected the margins from all batches of size 64 in the MSCOCO training data, where the data is randomly shuffled.

**Additional Experiments.** Here, we additionally compare the margin distribution of CLIP trained at temperature $\tau = 0.01$, without or with our regularization term. We could observe that the margin distribution shifts to the right with the regularization term, which alleviates the negative influence of an extremely small temperature value.

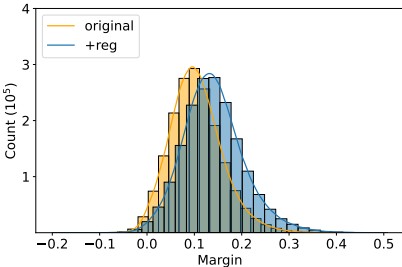

**Figure 6:** The distribution of the margins with regard to CLIP models trained $\tau = 0.01$ with or withour regularization. Margin is computed within each batch of the data.

## C.5 ZERO-SHOT TRANSFER AND LINEAR PROBING

**Setup.** In the evaluation of zero-shot transfer and linear probing, we use CC3M (Sharma et al., 2018) as the pre-training dataset, which contains around $3,318,332$ image-caption pairs gathered from the web. While some URLs are broken so that we cannot download the images, we eventually reached a pre-training dataset of $2,786,288$ data pairs. When training CLIP models, we use the default coefficients of CyCLIP regularization terms of $\lambda_1 = 0.25$ and $\lambda_2 = 0.25$. For our regularization term, we use a coefficient of $\lambda = 0.1$. As in CLIP, we set the temperature $\tau$ from $0.07$, equivalently having maximum logit scale at $2.6593$. Lastly, we use a training batch size of 32 and trained for 8 epochs in the results reported in section 7.2.

**Table 6:** Summary of datasets used for zero-shot transfer and linear probing.

| Dataset | Classes | Class Description |
|---|---|---|
| CIFAR10 | 10 | Categories of animals and vehicles |
| CIFAR100 | 100 | Categories of objects including animals, foods, vehicles and people |
| STL10 | 10 | Categories of animals and vehicles |
| Food101 | 101 | Categories of foods/dishes |
| ImageNetV2 | 1000 | Categories of objects including animals, foods, vehicles and people |
| DTD | 47 | Categories of textures |
| Flowers102 | 102 | Categories of flower species |
| Oxford-IIIT Pet | 37 | Categories of cats and dogs |

**Evaluations.** As similar in previous works (Radford et al., 2021; Yao et al., 2022; Mu et al., 2022; Goel et al., 2022), we consider the following image classification tasks for zero-shot transfer and linear probing: CIFAR10/100 (Krizhevsky, 2009), STL10 (Coates et al., 2011), Food101 (Bossard et al., 2014), ImageNetV2 (Recht et al., 2019), DTD (Describable Textures,Cimpoi et al. (2014)), Flowers102 (Nilsback & Zisserman, 2008) and Oxford-IIIT Pet (Parkhi et al., 2012). The dataset statistics are reported in Table 6. For zero-shot transfer, we use the same prompt engineering and ensembling as the original CLIP and report the top-1 accuracy. For linear probing, as the same in CLIP, we train a logistic regression classifier on the image embeddings generated by the image encoder of pre-trained CLIP models on the training data from the considered datasets. The classifiers are all trained to convergence and we report the test accuracy on each of the test dataset of the tasks. We note that, due to the limitation of the training data CC3M, the zero-shot test accuracy of all CLIP-objectives on Flowers102 and Oxford-IIIT Pet are near random guesses. Therefore, we omit these datasets for zero-shot transfer.

**Additional Experiments.** We additionally report the zero-shot transfer results of the original CLIP objective and adding our regularziation term, on a different visual encoder architecture of TinyViT (Wu et al., 2022) with pre-trained weights from Huggingface.

**Table 7:** Zero-shot top-1 accuracy (%). Notably, adding the regularization term successfully improves the baselines on 5 out of the 6 datasets.

| | CIFAR10 | CIFAR100 | STL10 | Food101 | ImageNetV2 | DTD | Average |
|---|---|---|---|---|---|---|---|
| CLIP | 52.02 | 15.57 | 81.89 | 7.92 | **16.91** | **11.80** | 31.02 |
| CLIP+Reg | **53.30** | **19.67** | **83.76** | **7.99** | 16.06 | 11.53 | **32.05** |

## D  PROOF OF RESULTS IN SECTION 3

*Proof of Theorem 3.3.* We first prove that $L_{S'}(f, \tau)$ is upper bounded by $4M \log B/\tau$.

$$L_{S'}(f, \tau) = \frac{1}{B} \sum_{i \in S'} \log \left( \sum_{j \in S'} \exp \left( [f(\mathbf{x}_j, \mathbf{y}_i) - f(\mathbf{x}_i, \mathbf{y}_i)]/\tau \right) \right)$$

$$+ \frac{1}{B} \sum_{i \in S'} \log \left( \sum_{j \in S'} \exp \left( [f(\mathbf{x}_i, \mathbf{y}_j) - f(\mathbf{x}_i, \mathbf{y}_i)]/\tau \right) \right)$$

$$\leq \frac{1}{B} \sum_{i \in S'} \log \left( \sum_{j \in S'} \exp \left( 2M/\tau \right) \right) + \frac{1}{B} \sum_{i \in S'} \log \left( \sum_{j \in S'} \exp \left( 2M/\tau \right) \right)$$

$$= 4M \log B/\tau. \tag{D.1}$$

where the inequality is by the fact the $|f| \leq M$. On the other hand, we have that

$$L_{S'}(f, \tau) = \frac{1}{B} \sum_{i \in S'} \log \left( \sum_{j \in S'} \exp \left( [f(\mathbf{x}_j, \mathbf{y}_i) - f(\mathbf{x}_i, \mathbf{y}_i)]/\tau \right) \right)$$

$$+ \frac{1}{B} \sum_{i \in S'} \log \left( \sum_{j \in S'} \exp \left( [f(\mathbf{x}_i, \mathbf{y}_j) - f(\mathbf{x}_i, \mathbf{y}_i)]/\tau \right) \right)$$

$$\geq \frac{2}{B} \sum_{i \in S'} \log \left( \exp \left( [f(\mathbf{x}_i, \mathbf{y}_i) - f(\mathbf{x}_i, \mathbf{y}_i)]/\tau \right) \right)$$

$$\geq 0.$$

where the inequality is because Exp function is greater than $0$. Therefore we have proved that $L_{S'}(f,\tau) \in (0, 4M \log(B)/\tau]$. For all $f_1, f_2 \in \mathcal{F}$ and any batch $S'$ with size $B$, we have that

$$
\begin{aligned}
L_{S'}(f_1,\tau) - L_{S'}(f_2,\tau) &= \frac{1}{B} \sum_{i \in S'} \log \left( \sum_{j \in S'} \exp\left([f_1(\mathbf{x}_j, \mathbf{y}_i) - f_1(\mathbf{x}_i, \mathbf{y}_i)]/\tau\right) \right) \\
&\quad - \frac{1}{B} \sum_{i \in S'} \log \left( \sum_{j \in S'} \exp\left([f_2(\mathbf{x}_j, \mathbf{y}_i) - f_2(\mathbf{x}_i, \mathbf{y}_i)]/\tau\right) \right) \\
&\quad + \frac{1}{B} \sum_{i \in S'} \log \left( \sum_{j \in S'} \exp\left([f_1(\mathbf{x}_i, \mathbf{y}_j) - f_1(\mathbf{x}_i, \mathbf{y}_i)]/\tau\right) \right) \\
&\quad - \frac{1}{B} \sum_{i \in S'} \log \left( \sum_{j \in S'} \exp\left([f_2(\mathbf{x}_i, \mathbf{y}_j) - f_2(\mathbf{x}_i, \mathbf{y}_i)]/\tau\right) \right) \\
&\leq \frac{1}{B} \sum_{i \in S'} \log \left( \sum_{j \in S'} \exp\left([f_1(\mathbf{x}_j, \mathbf{y}_i) - f_1(\mathbf{x}_i, \mathbf{y}_i)]/\tau\right) \right) \\
&\quad - \frac{1}{B} \sum_{i \in S'} \log \left( \sum_{j \in S'} \exp\left([f_1(\mathbf{x}_j, \mathbf{y}_i) - f_1(\mathbf{x}_i, \mathbf{y}_i) - 2\|f_1 - f_2\|_\infty]/\tau\right) \right) \\
&\quad + \frac{1}{B} \sum_{i \in S'} \log \left( \sum_{j \in S'} \exp\left([f_1(\mathbf{x}_i, \mathbf{y}_j) - f_1(\mathbf{x}_i, \mathbf{y}_i)]/\tau\right) \right) \\
&\quad - \frac{1}{B} \sum_{i \in S'} \log \left( \sum_{j \in S'} \exp\left([f_1(\mathbf{x}_i, \mathbf{y}_j) - f_1(\mathbf{x}_i, \mathbf{y}_i) - 2\|f_1 - f_2\|_\infty]/\tau\right) \right) \\
&= 4\|f_1 - f_2\|_\infty/\tau.
\end{aligned}
$$

Similarly, we can get another direction $L_{S'}(f_2,\tau) - L_{S'}(f_1,\tau) \leq 4\|f_1 - f_2\|_\infty/\tau$, which yields to $|L_{S'}(f_2,\tau) - L_{S'}(f_1,\tau)| \leq 4\|f_1 - f_2\|_\infty/\tau$. Taking the expectation gives that $|L_{\mathcal{D}^B}(f_2,\tau) - L_{\mathcal{D}^B}(f_1,\tau)| \leq 4\|f_1 - f_2\|_\infty/\tau$. By the definition of the covering set, the function class $\mathcal{F}$ can be covered by $K$ subsets $\mathcal{B}_1, \ldots, \mathcal{B}_K$, that is $\mathcal{F} = \mathcal{B}_1 \cup \ldots \cup \mathcal{B}_K$, where $K = \mathcal{N}(\mathcal{F}, \tau\epsilon/16)$ and $\mathcal{B}_1, \ldots \mathcal{B}_K$ are the balls of the radius $\tau \cdot \epsilon/16$ centered at $f_1, \ldots, f_K$. Then we have that

$$
\begin{aligned}
&\mathbb{P}_{S \sim \mathcal{D}^n} \left[ \sup_{f \in \mathcal{F}} \left| L_{\mathcal{D}^B}(f,\tau) - \widehat{L}_S(f,\tau) \right| \geq \epsilon \right] \\
&\leq \sum_{k \in [K]} \mathbb{P}_{S \sim \mathcal{D}^n} \left[ \sup_{f \in \mathcal{B}_k} \left| L_{\mathcal{D}^B}(f,\tau) - \widehat{L}_S(f,\tau) \right| \geq \epsilon \right] \\
&\leq \sum_{k \in [K]} \mathbb{P}_{S \sim \mathcal{D}^n} \left[ \left| L_{\mathcal{D}^B}(f_k,\tau) - \widehat{L}_S(f_k,\tau) \right| \geq \epsilon/2 \right] \\
&= \sum_{k \in [K]} \mathbb{P}_{S \sim \mathcal{D}^n} \left[ \left| L_{\mathcal{D}^B}(f_k,\tau) - (1/n) \sum_{i \in [n]} L_{S_i}(f_k,\tau) \right| \geq \epsilon/2 \right] \\
&\leq 2K \exp \left( -\frac{n\epsilon^2\tau}{8M \log B} \right) \\
&= 2\mathcal{N}(\mathcal{F}, \tau\epsilon/16) \exp \left( -\frac{n\epsilon^2\tau}{8M \log B} \right),
\end{aligned}
\tag{D.2}
$$

the first inequality is by union bound, the second is by triangle inequality, and the third is by Hoeffding's inequality and (D.1). Finally, plugging the condition $n \geq (8\tau^{-1}\epsilon^{-2}M \log B) \log(2\mathcal{N}(\mathcal{F}, \epsilon/8M)/\delta)$ into (D.2) we have that

$$
\mathbb{P}_{S \sim \mathcal{D}^n} \left[ \sup_{f \in \mathcal{F}} \left| L_{\mathcal{D}^B}(f,\tau) - \widehat{L}_S(f,\tau) \right| \geq \epsilon \right] \leq \delta,
$$

which completes the proof. $\qquad\square$

# E    PROOF OF RESULTS IN SECTION 4

**Lemma E.1.** For $b_j \geq 0, j \in [m]$, we have that

$$\log \left( 1 + \sum_{j \in [m]} b_j \right) \leq \sum_{j \in [m]} \log(1 + b_j).$$

*Proof.* Notice that

$$\Pi_{j \in [J]}(1 + b_j) \geq 1 + \sum_{j \in [J]} b_j.$$

Taking the logarithm over both sides completes the proof.  □

**Lemma E.2.** Suppose that $a_1, \ldots a_m$ are i.i.d random variable sample lies in $[-R, R]$ where $R \geq 1$, with mean $\mu := \mathbb{E}[a_1]$ and variance $\sigma^2 := \mathbb{E}[(a_1 - \mathbb{E}[a_1])^2]$. Then we have that

$$\mathbb{E}[\log \left( \sum_{i=1}^{m} \exp(a) \right) \geq \log(m) + \mu + \frac{m-1}{4mR^2} \sigma^2.$$

*Proof.* Let $\bar{a} = \left[ \sum_{i=1}^{m} a_i \right]/m$

$$\log \left( \sum_{i=1}^{m} \exp(a_i) \right) = \log(m) + \frac{1}{m} \sum_{i=1}^{m} a_i + \log \left( \frac{1}{m} \sum_{i=1}^{m} \exp(a - \bar{a}) \right)$$

$$\geq \log(m) + \frac{1}{m} \sum_{i=1}^{m} a_i + \log \left( 1 + \frac{1}{3mR^2} \sum_{i=1}^{m} [a - \bar{a}]^2 \right)$$

$$\geq \log(m) + \frac{1}{m} \sum_{i=1}^{m} a_i + \frac{1}{4mR^2} \sum_{i=1}^{m} [a - \bar{a}]^2.$$

where the first inequality is by $\exp(t) \geq 1 + t + t^2/(3R^2), \forall t \in [-R, R]$, the second inequality is due to $\log(1 + t) \geq 3t/4, \forall t \in [0, 1/3]$.  □

**Lemma E.3.** Suppose $f^*$ is the function that satisfies Assumption 4.1, then we have that

$$L_{\mathcal{D}^B}(f^*, \tau) \leq 2\mathbb{E} \left[ \log \left( \sum_{t \in [B]} \mathbb{1}(\mathbf{z}_t = \mathbf{z}_1) \right) \right] + 6MB\alpha/\tau + 3\sqrt[3]{6MB\beta}/\tau + 2B \exp(-\gamma/\tau)$$

*Proof.* Let the event $\mathcal{E}_t$ be the case that either i) $\mathbf{z}_t = \mathbf{z}_1$ and $|f^*(\mathbf{x}_t, \mathbf{y}_1) - f^*(\mathbf{x}_1, \mathbf{y}_1)| \leq \rho$ or ii) $\mathbf{z}_t \neq \mathbf{z}_1$ and $f^*(\mathbf{x}_t, \mathbf{y}_1) - f^*(\mathbf{x}_1, \mathbf{y}_1) \leq -\gamma$. We also denote the complementary set of $\mathcal{E}_t$ to be $\mathcal{E}_t^c$. By Assumption 4.1, we have that

$$\mathbb{P}(\mathcal{E}_t, \mathbf{z}_t = \mathbf{z}_1) \leq \beta/\rho^2$$
$$\mathbb{P}(\mathcal{E}_t, \mathbf{z}_t \neq \mathbf{z}_1) \leq \alpha.$$

the first inequality is by Chebyshev's inequality, and the second is by margin assumption. Therefore, we have that $\mathbb{P}(\mathcal{E}_t^c) \leq \alpha + \beta/\rho^2$. Next, let us decompose $L_{\mathcal{D}^B}(f^*, \tau)$ into three parts,

$$L_{\mathcal{D}^B}(f^*, \tau) = \mathbb{E} \left[ \log \left( \sum_{t \in [B]} \mathbb{1}(\mathbf{z}_t \neq \mathbf{z}_1) \mathbb{1}(\mathcal{E}_t) \exp \left( [f^*(\mathbf{x}_1, \mathbf{y}_t) - f^*(\mathbf{x}_1, \mathbf{y}_1)]/\tau \right) \right. \right.$$

$$+ \sum_{t \in [B]} \mathbb{1}(\mathcal{E}_t^c) \exp \left( [f^*(\mathbf{x}_1, \mathbf{y}_t) - f^*(\mathbf{x}_1, \mathbf{y}_1)]/\tau \right)$$

$$\left. \left. + \sum_{t \in [B]} \mathbb{1}(\mathbf{z}_t = \mathbf{z}_1) \mathbb{1}(\mathcal{E}_t) \exp \left( [f^*(\mathbf{x}_1, \mathbf{y}_t) - f^*(\mathbf{x}_1, \mathbf{y}_1)]/\tau \right) \right) \right]$$

$$+ \mathbb{E} \left[ \log \left( \sum_{t \in [B]} \mathbb{1}(\mathbf{z}_t \neq \mathbf{z}_1) \mathbb{1}(\mathcal{E}_t) \exp \left( [f^*(\mathbf{x}_t, \mathbf{y}_1) - f^*(\mathbf{x}_1, \mathbf{y}_1)]/\tau \right) \right. \right.$$

$$+ \sum_{t \in [B]} \mathbb{1}(\mathcal{E}_t^c) \exp\left( \left[ f^*(\mathbf{x}_t, \mathbf{y}_1) - f^*(\mathbf{x}_1, \mathbf{y}_1) \right]/\tau \right)$$

$$+ \left. \sum_{t \in [B]} \mathbb{1}(\mathbf{z}_t = \mathbf{z}_1)\, \mathbb{1}(\mathcal{E}_t) \exp\left( \left[ f^*(\mathbf{x}_t, \mathbf{y}_1) - f^*(\mathbf{x}_1, \mathbf{y}_1) \right]/\tau \right) \right) \right]$$

$$\leq 2\mathbb{E}\left[ \log\left( 1 + B\exp\left(-\gamma/\tau\right) + \sum_{t \geq 2} \mathbb{1}(\mathcal{E}_t^c) \exp\left(2M/\tau\right) + \sum_{t \geq 2} \mathbb{1}(\mathbf{z}_t = \mathbf{z}_1) \exp\left(\rho/\tau\right) \right) \right]$$

$$\leq 2\underbrace{\mathbb{E}\left[ \log\left( 1 + B\exp\left(-\gamma/\tau\right) \right) \right]}_{I_1} + \sum_{t \geq 2} 2\underbrace{\mathbb{E}\left[ \log\left( 1 + \mathbb{1}(\mathcal{E}_t^c) \exp\left(2M/\tau\right) \right) \right]}_{I_2}$$

$$+ 2\underbrace{\mathbb{E}\left[ \log\left( 1 + \sum_{t \geq 2} \mathbb{1}(\mathbf{z}_t = \mathbf{z}_1) \exp\left(\rho/\tau\right) \right) \right]}_{I_3} \tag{E.1}$$

where the first inequality is by Assumption 4.1, the second inequality is due to Lemma E.1. Next, we will bound $I_1, I_2, I_3$ separately.

$$I_1 \leq B\exp(-\gamma/\tau), \tag{E.2}$$

where the inequality is due to the fact that $\log(1 + x) \leq x$.

$$I_2 = \mathbb{E}\left[ \mathbb{1}(\mathcal{E}_t^c) \log\left( 1 + \exp\left(2M/\tau\right) \right) \right] \leq \mathbb{P}(\mathcal{E}_t^c)\frac{3M}{\tau} = (\alpha + \beta/\rho^2) \cdot \frac{3M}{\tau}. \tag{E.3}$$

where the first equality is due to $\log\left( 1 + \mathbb{1}(\mathcal{E}_t^c) \exp\left(2M/\tau\right) \right) = 0)$ when $\mathbb{1}(\mathcal{E}_t^c) = 0$, the first inequality is due to $\log\left( 1 + \exp\left(2M/\tau\right) \right) \leq 3M/\tau$. The last inequality is due to $\mathbb{P}(\mathcal{E}_t^c) \leq \alpha + \beta/\rho^2$.

$$I_3 \leq \mathbb{E}\left[ \log\left( \exp\left(\rho/\tau\right) + \sum_{t \geq 2} \mathbb{1}(\mathbf{z}_t = \mathbf{z}_1) \exp\left(\rho/\tau\right) \right) \right]$$

$$= \rho/\tau + \mathbb{E}\left[ \log\left( \sum_{t \in [B]} \mathbb{1}(\mathbf{z}_t = \mathbf{z}_1) \right) \right]. \tag{E.4}$$

where the inequality is because $1 \leq \exp(\rho/\tau)$.
Plugging (E.2), (E.3) and (E.4) into (E.1) gives that,

$$L_{\mathcal{D}^B}(f^*, \tau) \leq 2B\exp(-\gamma/\tau) + 6MB\alpha/\tau + 6MB\beta/(\tau\rho^2) + 2\rho/\tau + 2\mathbb{E}\left[ \log\left( \sum_{t \in [B]} \mathbb{1}(\mathbf{z}_t = \mathbf{z}_1) \right) \right]$$

$$\leq 2\mathbb{E}\left[ \log\left( \sum_{t \in [B]} \mathbb{1}(\mathbf{z}_t = \mathbf{z}_1) \right) \right] + 6MB\alpha/\tau + 3\sqrt[3]{6MB\beta}/\tau + 2B\exp(-\gamma/\tau),$$

where the second inequality is by choosing $\rho = \sqrt[3]{6MB\beta}$. $\qquad\square$

*Proof of Theorem 4.2.* First by Lemma E.3, we have that

$$L_{\mathcal{D}^B}(\widehat{f}, \tau) \leq L_{\mathcal{D}^B}(f^*, \tau) + \epsilon \leq 2\mathbb{E}\left[ \log\left( \sum_{t \in [B]} \mathbb{1}(\mathbf{z}_t = \mathbf{z}_1) \right) \right] + \epsilon' \tag{E.5}$$

where $\epsilon' = \epsilon + 6MB\alpha/\tau + 3\sqrt[3]{6MB\beta}/\tau + 2B\exp(-\gamma/\tau)$. Notice that

$$L_{\mathcal{D}^B}(\widehat{f}, \tau) = \underbrace{\mathbb{E}\left[ \log\left( \sum_{t \in [B]} \exp\left( \left[\widehat{f}(\mathbf{x}_1, \mathbf{y}_t) - \widehat{f}(\mathbf{x}_1, \mathbf{y}_1)\right]/\tau \right) \right) \right]}_{I_1}$$

$$+ \mathbb{E}\left[\log\left(\sum_{t\in[B]}\exp\left([\widehat{f}(\mathbf{x}_t,\mathbf{y}_1) - \widehat{f}(\mathbf{x}_1,\mathbf{y}_1)]/\tau\right)\right)\right] \tag{E.6}$$

$$\underbrace{\phantom{+ \mathbb{E}\left[\log\left(\sum_{t\in[B]}\exp\left([\widehat{f}(\mathbf{x}_t,\mathbf{y}_1) - \widehat{f}(\mathbf{x}_1,\mathbf{y}_1)]/\tau\right)\right)\right]}}_{I_2}$$

Next, we prove the bullets in Theorem 4.2 one by one.

**First and Second Bullet in Theorem 4.2:** Denote the event $\mathcal{E}$ as the case that for all $t \geq 1$, $\mathbf{z}_t \neq \mathbf{z}_1$, which is the event that CLIP favored. We first lower bound $I_1$.

$$
\begin{aligned}
I_1 &= \mathbb{E}\Bigg[\log\Bigg(\sum_{t\in[B]}\mathbb{1}(\mathbf{z}_t\neq\mathbf{z}_1)\exp\left([\widehat{f}(\mathbf{x}_t,\mathbf{y}_1)-\widehat{f}(\mathbf{x}_1,\mathbf{y}_1)]/\tau\right)\\
&\quad +\sum_{t\in[B]}\mathbb{1}(\mathbf{z}_t=\mathbf{z}_1)\exp\left([\widehat{f}(\mathbf{x}_t,\mathbf{y}_1)-\widehat{f}(\mathbf{x}_1,\mathbf{y}_1)]/\tau\right)\Bigg)\Bigg]\\
&= \mathbb{E}\Bigg[\log\Bigg(\sum_{t\in[B]}\mathbb{1}(\mathbf{z}_t\neq\mathbf{z}_1)\exp\left([\widehat{f}(\mathbf{x}_t,\mathbf{y}_1)-\widehat{f}(\mathbf{x}_1,\mathbf{y}_1)]/\tau\right)\\
&\quad +\sum_{t\in[B]}\mathbb{1}(\mathbf{z}_t=\mathbf{z}_1)\exp\left([\widehat{f}(\mathbf{x}_t,\mathbf{y}_1)-\widehat{f}(\mathbf{x}_1,\mathbf{y}_1)]/\tau\right)\Bigg)\Bigg]\\
&\geq \mathbb{E}\Bigg[\mathbb{1}(\mathcal{E})\log\Bigg(\sum_{t\in[B]}\mathbb{1}(\mathbf{z}_t\neq\mathbf{z}_1)\exp\left([\widehat{f}(\mathbf{x}_t,\mathbf{y}_1)-\widehat{f}(\mathbf{x}_1,\mathbf{y}_1)]/\tau\right)+1\Bigg)\Bigg]\\
&\quad +\mathbb{E}\Bigg[\mathbb{1}(\mathcal{E}^c)\log\Bigg(\sum_{t\in[B]}\mathbb{1}(\mathbf{z}_t=\mathbf{z}_1)\exp\left([\widehat{f}(\mathbf{x}_t,\mathbf{y}_1)-\widehat{f}(\mathbf{x}_1,\mathbf{y}_1)]/\tau\right)\Bigg)\Bigg]\\
&= \mathbb{E}\Bigg[\mathbb{1}(\mathcal{E})\log\Bigg(\sum_{t\in[B]}\mathbb{1}(\mathbf{z}_t\neq\mathbf{z}_1)\exp\left([\widehat{f}(\mathbf{x}_t,\mathbf{y}_1)-\widehat{f}(\mathbf{x}_1,\mathbf{y}_1)]/\tau\right)+1\Bigg)\Bigg]\\
&\quad +\mathbb{E}\Bigg[\log\Bigg(\sum_{t\in[B]}\mathbb{1}(\mathbf{z}_t=\mathbf{z}_1)\exp\left([\widehat{f}(\mathbf{x}_t,\mathbf{y}_1)-\widehat{f}(\mathbf{x}_1,\mathbf{y}_1)]/\tau\right)\Bigg)\Bigg]\\
&\geq \mathbb{E}\Bigg[\mathbb{1}(\mathcal{E})\log\Bigg(\sum_{t\in[B]}\mathbb{1}(\mathbf{z}_t\neq\mathbf{z}_1)\exp\left([\widehat{f}(\mathbf{x}_t,\mathbf{y}_1)-\widehat{f}(\mathbf{x}_1,\mathbf{y}_1)]/\tau\right)+1\Bigg)\Bigg]\\
&\quad +\mathbb{E}\Bigg[\log\Bigg(\sum_{t\in[B]}\mathbb{1}(\mathbf{z}_t=\mathbf{z}_1)\exp\left(\mathbb{E}[\widehat{f}(\mathbf{x}_t,\mathbf{y}_1)-\widehat{f}(\mathbf{x}_1,\mathbf{y}_1)|\mathbf{z}_t,\mathbf{z}_1]/\tau\right)\Bigg)\Bigg]\\
&= \mathbb{E}\Bigg[\mathbb{1}(\mathcal{E})\log\Bigg(\sum_{t\in[B]}\mathbb{1}(\mathbf{z}_t\neq\mathbf{z}_1)\exp\left([\widehat{f}(\mathbf{x}_t,\mathbf{y}_1)-\widehat{f}(\mathbf{x}_1,\mathbf{y}_1)]/\tau\right)+1\Bigg)\Bigg]\\
&\quad +\mathbb{E}\Bigg[\log\Big(\big|\{t\in[B]\big|\mathbf{z}_t=\mathbf{z}_1\}\big|\Big)\Bigg].
\end{aligned}
\tag{E.7}
$$

where the first inequality is because when $\mathcal{E}$ holds $\sum_{t\in[B]}\mathbb{1}(\mathbf{z}_t = \mathbf{z}_1)\exp\left([\widehat{f}(\mathbf{x}_t,\mathbf{y}_1) - \widehat{f}(\mathbf{x}_1,\mathbf{y}_1)]/\tau\right) = 1$ when $\mathcal{E}^c$ holds $\sum_{t\in[B]}\mathbb{1}(\mathbf{z}_t\neq\mathbf{z}_1)\exp\left([\widehat{f}(\mathbf{x}_t,\mathbf{y}_1)-\widehat{f}(\mathbf{x}_1,\mathbf{y}_1)]/\tau\right) \geq 0$, the last second equality is because when $\mathcal{E}$ holds $\sum_{t\in[B]}\mathbb{1}(\mathbf{z}_t=\mathbf{z}_1)\exp\left([\widehat{f}(\mathbf{x}_t,\mathbf{y}_1)-\widehat{f}(\mathbf{x}_1,\mathbf{y}_1)]/\tau\right) = 1$, the second inequality is because LogSumExp function is convex, and the last equality is due to $\mathbb{E}[[\widehat{f}(\mathbf{x}_t,\mathbf{y}_1)-\widehat{f}(\mathbf{x}_1,\mathbf{y}_1)]|\mathbf{z}_t,\mathbf{z}_1] = 0$ when $\mathbf{z}_t = \mathbf{z}_1$. Similarly, we can prove

$$
\begin{aligned}
I_2 &\geq \mathbb{E}\Bigg[\mathbb{1}(\mathcal{E})\log\Bigg(\sum_{t\in[B]}\mathbb{1}(\mathbf{z}_t\neq\mathbf{z}_1)\exp\left([\widehat{f}(\mathbf{x}_1,\mathbf{y}_t)-\widehat{f}(\mathbf{x}_1,\mathbf{y}_1)]/\tau\right)+1\Bigg)\Bigg]\\
&\quad +\mathbb{E}\Bigg[\log\Big(\big|\{t\in[B]\big|\mathbf{z}_t=\mathbf{z}_1\}\big|\Big)\Bigg].
\end{aligned}
\tag{E.8}
$$

Notice that when event $\mathcal{E}$ holds, $\mathbf{z}_t \neq \mathbf{z}_1$ holds for all $t \geq 2$. Therefore, plugging the (E.7) and (E.8) into (E.6) gives,

$$\mathbb{E}\left[ \mathbb{1}(\mathcal{E}) \log \left( \sum_{t \geq 2} \exp \left( [\widehat{f}(\mathbf{x}_t, \mathbf{y}_1) - \widehat{f}(\mathbf{x}_1, \mathbf{y}_1)]/\tau \right) + 1 \right) \right] \leq \epsilon' \tag{E.9}$$

$$\mathbb{E}\left[ \mathbb{1}(\mathcal{E}) \log \left( \sum_{t \geq 2} \exp \left( [\widehat{f}(\mathbf{x}_1, \mathbf{y}_t) - \widehat{f}(\mathbf{x}_1, \mathbf{y}_1)]/\tau \right) + 1 \right) \right] \leq \epsilon'. \tag{E.10}$$

$$\tag{E.11}$$

Let us compute the probability of $\mathcal{E}$ given $\mathbf{z}_1$. Let $\mathbf{z}_1 = \mathbf{v}_1$ without loss of generality, we have that

$$\mathbb{P}(\mathcal{E}|\mathbf{z} = \mathbf{v}_1) = (1 - p_1)^{B-1}.$$

Therefore $\mathbb{P}(\mathcal{E}|\mathbf{z} = \mathbf{v}_1)$ is always positive and is greater than $1/2$ as long as $B \leq 1/p_1$.
Next, consider the following situation. Given $\mathbf{z}_1 = \mathbf{v}_1$, we generate sequence $\mathbf{z}'_1, \ldots, \mathbf{z}'_L$ with length $L = \lceil \log(2K)/(B-1) \min p_k \rceil (B-1)$, such that each $\mathbf{z}'_1, \ldots, \mathbf{z}'_L$ are generated from $\mathcal{D}_{\mathbf{z}|\mathbf{z} \neq \mathbf{v}_1}$. The probability that the sequence includes $\mathbf{v}_k$ is

$$1 - (1 - p_k/(1 - p_k))^L \geq 1 - (1 - p_k)^L \geq 1 - \exp(-Lp_k) \geq 1 - \exp(-L \min p_k).$$

Therefore the probability that the sequence can cover all the other $K - 1$ classes is at least

$$1 - K \exp(-L \min p_k) \geq 1/2.$$

Then we look deeper into

$$\mathbb{E}\left[ \log \left( \sum_{t \geq 2} \exp \left( [\widehat{f}(\mathbf{x}_t, \mathbf{y}_1) - \widehat{f}(\mathbf{x}_1, \mathbf{y}_1)]/\tau \right) + 1 \right) \Big| \mathbf{z}_1 = \mathbf{v}_1, \mathbf{z}_2 \neq \mathbf{v}_1, \ldots, \mathbf{z}_K \neq \mathbf{v}_1 \right].$$

We can introduce $L/(B-1)$ copies $\mathbf{x}_t^{(l)}$ with $l \in [L/(B-1)]$ for $t \geq 2$, then we have that

$$\left( L/(B-1) \right) \cdot \mathbb{E}\left[ \log \left( \sum_{t \geq 2} \exp \left( [\widehat{f}(\mathbf{x}_t, \mathbf{y}_1) - \widehat{f}(\mathbf{x}_1, \mathbf{y}_1)]/\tau \right) + 1 \right) \Big| \mathbf{z}_1 = \mathbf{v}_1, \mathbf{z}_2 \neq \mathbf{v}_1, \ldots, \mathbf{z}_K \neq \mathbf{v}_1 \right]$$

$$= \mathbb{E}\left[ \sum_l \log \left( \sum_{t \geq 2} \exp \left( [\widehat{f}(\mathbf{x}_t^{(l)}, \mathbf{y}_1) - \widehat{f}(\mathbf{x}_1, \mathbf{y}_1)]/\tau \right) + 1 \right) \Big| \mathbf{z}_1 = \mathbf{v}_1, \mathbf{z}_2^{(l)}, \ldots, \mathbf{z}_K^{(l)} \neq \mathbf{v}_1 \right]$$

$$\geq \mathbb{E}\left[ \log \left( \sum_l \sum_{t \geq 2} \exp \left( [\widehat{f}(\mathbf{x}_t^{(l)}, \mathbf{y}_1) - \widehat{f}(\mathbf{x}_1, \mathbf{y}_1)]/\tau \right) + 1 \right) \Big| \mathbf{z}_1 = \mathbf{v}_1, \mathbf{z}_2^{(l)}, \ldots, \mathbf{z}_K^{(l)} \neq \mathbf{v}_1 \right]$$

$$\geq \mathbb{E}\left[ \log \left( \sum_{k \in [K]} \exp \left( [\widehat{f}(\mathbf{x}_k, \mathbf{y}) - \widehat{f}(\mathbf{x}^*, \mathbf{y})]/\tau \right) \right) \Big| \mathbf{z} = \mathbf{v}_1 \right]. \tag{E.12}$$

where the first inequality is by Lemma E.1, the second inequality is by the fact that the Exp function is greater than 0, and the $\mathbf{x}_k, \mathbf{x}^*$ in the last line are the ones that defined in Theorem 4.2. Plugging (E.12) into (E.9) and applying total expectation completes the proof for the second bullet. The proof for the first bullet is the same.

**Third Bullet in Theorem 4.2:** By the third equality in (E.7), we have that

$$I_1 \geq \mathbb{E}\left[ \log \left( \sum_{t \in [B]} \mathbb{1}(\mathbf{z}_t = \mathbf{z}_1) \exp \left( [\widehat{f}(\mathbf{x}_t, \mathbf{y}_1) - \widehat{f}(\mathbf{x}_1, \mathbf{y}_1)]/\tau \right) \right) \right]$$

$$= \mathbb{E}\left[ \mathbb{E}\left[ \log \left( \sum_{t \in [B]} \mathbb{1}(\mathbf{z}_t = \mathbf{z}_1) \exp \left( \widehat{f}(\mathbf{x}_t, \mathbf{y}_1)/\tau \right) \right) \Big| \mathbf{z}_1, \ldots, \mathbf{z}_B \right] \right] - \mathbb{E}[\widehat{f}(\mathbf{x}_1, \mathbf{y}_1)/\tau]$$

$$\geq \mathbb{E}\left[ \log \left( \left| \left\{ t \in [B] \Big| \mathbf{z}_t = \mathbf{z}_1 \right\} \right| \right) \right] + \mathbb{E}\left[ \frac{\left| \left\{ t \in [B] \Big| \mathbf{z}_t = \mathbf{z}_1 \right\} \right| - 1}{4M^2 \left| \left\{ t \in [B] \Big| \mathbf{z}_t = \mathbf{z}_1 \right\} \right|} \mathrm{Var}_{\mathbf{x}_1|\mathbf{z}_1}(\widehat{f}(\mathbf{x}_1, \mathbf{y}_1)) \right].$$

$$\tag{E.13}$$

where the inequality is by Lemma E.2. Next we will We analyze the distribution of $\left\{t \in [B]\middle| \mathbf{z}_t = \mathbf{z}_1\right\}$. Without loss of generality, fix $\mathbf{z}_1 = \mathbf{v}_1$. We know that the probability that $\left\{t \in [B]\middle| \mathbf{z}_t = \mathbf{z}_1\right\} \geq 2$ is

$$1 - \mathbb{P}(\mathbf{z}_2 \neq \mathbf{z}_1) \cdot \ldots \cdot \mathbb{P}(\mathbf{z}_B \neq \mathbf{z}_1) \geq 1 - (1 - \min p_k)^{B-1} \geq \min\{0.25 * \min p_k \cdot (B - 1), 0.25\},$$

the last inequality holds since the strictly increasing function $F(s) = 1 - (1 - \min p_k)^s$ is 0 at $s = 0$ and have derivative lower bounded by 0.25 when $s \leq 1/\min p_k$. Therefore we can further lower bound (E.13) as follows,

$$I_1 \geq \mathbb{E}\left[\log\left(\left|\left\{t \in [B]\middle|\mathbf{z}_t = \mathbf{z}_1\right\}\right|\right)\right] + \mathbb{E}\left[\frac{\min\{0.25 * \min p_k \cdot (B - 1), 0.25\}}{8M^2}\mathrm{Var}_{\mathbf{x}_1|\mathbf{z}_1}(\widehat{f}(\mathbf{x}_1, \mathbf{y}_1))\right]$$

Similarly, we can prove that

$$I_2 \geq \mathbb{E}\left[\log\left(\left|\left\{t \in [B]\middle|\mathbf{z}_t = \mathbf{z}_1\right\}\right|\right)\right] + \mathbb{E}\left[\frac{\min\{0.25 * \min p_k \cdot (B - 1), 0.25\}}{8M^2}\mathrm{Var}_{\mathbf{y}_1|\mathbf{z}_1}(\widehat{f}(\mathbf{x}_1, \mathbf{y}_1))\right].$$

Plugging the bound of $I_1, I_2$ into (E.6) completes the proof for the third bullet of Theorem 4.2. □

## F  PROOF OF THE RESULTS IN SECTION 5

*Proof of Corollary 5.1.* For $(\mathbf{x}, \mathbf{z}) \sim \mathcal{D}_{\mathbf{x} \times \mathbf{z}}$, $\{\mathbf{y}_k \sim \mathcal{D}_{\mathbf{y}|\mathbf{v}_k}, k \in [K]\}$, let $\mathbf{y}^* = \sum_{k \in [K]} \mathbb{1}(\mathbf{z} = \mathbf{v}_k)\mathbf{y}_k$. Denote $\mathcal{E}$ to be the event that the top-r choice gives the wrong prediction. Then we have that,

$$\begin{aligned}
\epsilon' &\geq \mathbb{E}\left[\log\left(\sum_{k \in [K]}\exp\left([\widehat{f}(\mathbf{x}, \mathbf{y}_k) - \widehat{f}(\mathbf{x}, \mathbf{y}^*)]/\tau\right)\right)\right] \\
&\geq \mathbb{E}\left[\mathbb{1}(\mathcal{E})\log\left(\sum_{k \in [K]}\exp\left([\widehat{f}(\mathbf{x}, \mathbf{y}_k) - \widehat{f}(\mathbf{x}, \mathbf{y}^*)]/\tau\right)\right)\right] \\
&\geq \mathbb{E}\left[\mathbb{1}(\mathcal{E})\log(1 + r)\right] \\
&= \mathbb{P}(\mathcal{E})\log(1 + r),
\end{aligned}$$

where the first inequality is by the first bullet of Theorem 4.2, the second inequality is due to the fact that $\log\left(\sum_{k \in [K]}\exp\left([\widehat{f}(\mathbf{x}, \mathbf{y}_k) - \widehat{f}(\mathbf{x}, \mathbf{y}^*)]/\tau\right)\right) > 0$, the last inequality is due to $\log\left(\sum_{k \in [K]}\exp\left([\widehat{f}(\mathbf{x}, \mathbf{y}_k) - \widehat{f}(\mathbf{x}, \mathbf{y}^*)]/\tau\right)\right) \geq \log(1 + r)$ since there are at least $r + 1$ number of $\widehat{f}(\mathbf{x}, \mathbf{y}_k)$ are greater than $\widehat{f}(\mathbf{x}, \mathbf{y}^*)$ if the prediction is wrong. Therefore, we have that $\mathbb{P}(\mathcal{E}) \leq \epsilon'/\log(1 + r)$ which completes the proof. □

**Discussion for out-of-distribution zero shot learning.** The result in Corollary 5.1 can be generalized to out-of-distribution zero-shot transfer learning. For example, we can deal with the case where the distribution of the prompts $\mathcal{D}_{\mathbf{y}|\mathbf{v}_k}$ and the image distribution $\mathcal{D}_{\mathbf{x}}$ are shifted. In particular, let us consider the case that the distribution of the prompts is shifted to $\mathcal{D}'_{\mathbf{y}|\mathbf{v}_k}$ and the image distribution $\mathcal{D}_{\mathbf{x}}$ is shifted to $\mathcal{D}'_{\mathbf{x}}$. Then the original joint cumulative distribution function function $P(\mathbf{x}, \mathbf{z}, \mathbf{y}_1, \ldots, \mathbf{y}_K)$ is shifted to $Q(\mathbf{x}, \mathbf{z}, \mathbf{y}_1, \ldots, \mathbf{y}_K)$. Suppose $Q$ is absolutely continuous with respect to $P$, and the Pearson $\chi^2$ distance is bounded

$$\int\left(\frac{dQ}{dP} - 1\right)^2 dP \leq C.$$

Then we have that

$$\int\sqrt{\log\left(\sum_{k \in [K]}\exp\left([\widehat{f}(\mathbf{x}, \mathbf{y}_k) - \widehat{f}(\mathbf{x}, \mathbf{y}^*)]/\tau\right)\right)}dQ$$

$$= \int \sqrt{\log \Big( \sum_{k \in [K]} \exp \big( [\widehat{f}(\mathbf{x}, \mathbf{y}_k) - \widehat{f}(\mathbf{x}, \mathbf{y}^*)]/\tau \big) \Big)} \Big( \frac{dQ}{dP} \Big) dP$$

$$\leq \sqrt{\int \log \Big( \sum_{k \in [K]} \exp \big( [\widehat{f}(\mathbf{x}, \mathbf{y}_k) - \widehat{f}(\mathbf{x}, \mathbf{y}^*)]/\tau \big) \Big) dP} \cdot \sqrt{\int \Big( \frac{dQ}{dP} \Big)^2 dP}$$

$$= \sqrt{(C+1)\epsilon'},$$

where the first inequality is by Cauchy Schwartz inequality and the last equality is due to $\int \Big( \frac{dQ}{dP} \Big)^2 dP = \int \Big( \frac{dQ}{dP} - 1 \Big)^2 dP + 1 = C + 1$. Then we can follow a similar analysis in the proof of Corollary 5.1 and have that top-r test error is smaller than $\sqrt{(C+1)\epsilon'/\log(1+r)}$. There-fore, if the $\chi^2$ distance between the shifted distributions is bounded, we can still provide a top-$r$ error guarantee. It is worth noting the bound for out-of-distribution zero-shot learning is looser. If we want to do a more general zero shot analysis, we may need to add more data structure in Assumption 4.1.

*Proof of Lemma 5.4.* We can construct $\mathbf{W}^* = \mathbf{H}(\mathbf{H}^\top \mathbf{H})^{-1} \mathbf{P}(\mathbf{G}^\top \mathbf{G})^{-1} \mathbf{G}^\top$, where $\mathbf{P} \in \mathbb{R}^{(K_1+K_2) \times (K_1+K_3)}$ is the projection matrix $\begin{bmatrix} \mathbf{I} & \mathbf{0} \\ \mathbf{0} & \mathbf{0} \end{bmatrix}$ with rank $K_1$.

It is easy to verify that $\mathbf{H}^\top \mathbf{W}^* \mathbf{G} = \mathbf{P}$. Therefore we have that

$$\langle \mathbf{W}^* \mathbf{x}, \mathbf{y}' \rangle = \langle \mathbf{z}, \mathbf{z}' \rangle.$$

Then applying $\|\mathbf{v}_k\|_2 = 1$, $\langle \mathbf{v}_k, \mathbf{v}'_k \rangle \leq 1 - \gamma, \forall k \neq k'$ completes the proof . $\qquad \square$

**Lemma F.1.** $\|\nabla L_S(f_{\mathbf{W}}, \tau)\|_F \leq L$ where $L = 2\tau^{-1} \|\mathbf{G}\|_2 \|\mathbf{H}\|_2 (R^2 + 1)$.

*Proof.* First, we have that

$$\|\nabla_{\mathbf{W}} \langle \mathbf{W} \mathbf{x}, \mathbf{y} \rangle\|_F = \|\mathbf{x} \mathbf{y}^\top\|_F \leq \|\mathbf{x}\|_2 \|\mathbf{y}\|_2 \leq \|\mathbf{G}\|_2 \|\mathbf{H}\|_2 (R^2 + 1).$$

Therefore we have that $\|\nabla L_S(f_{\mathbf{W}}, \tau)\|_F \leq 2\tau^{-1} \|\mathbf{G}\|_2 \|\mathbf{H}\|_2 (R^2 + 1)$ since LogSumExp function is an 1-Lipschitz function. $\qquad \square$

*Proof of Theorem 5.5.* By the gradient update rule, we have that

$$\|\mathbf{W}^{(t)} - \mathbf{W}^*\|_F^2 - \|\mathbf{W}^{(t+1)} - \mathbf{W}^*\|_F^2$$
$$= 2\eta \langle \nabla \widehat{L}_S(\mathbf{W}^{(t)}, \tau), \mathbf{W}^{(t)} - \mathbf{W}^* \rangle - \eta^2 \|\nabla \widehat{L}_S(\mathbf{W}^{(t)}, \tau)\|_F^2$$
$$\geq 2\eta \widehat{L}_S(\mathbf{W}^{(t)}, \tau) - 2\eta \widehat{L}_S(\mathbf{W}^*, \tau) - \eta^2 L^2. \tag{F.1}$$

Take the telescope sum of (F.1) from 0 to $T - 1$ we have that

$$\frac{\sum_{t=0}^{T-1} \widehat{L}_S(\mathbf{W}^{(t)}, \tau)}{T} \leq \widehat{L}_S(\mathbf{W}^*, \tau) + \eta L^2 + \frac{\|\mathbf{W}^{(0)} - \mathbf{W}^*\|_F^2 - \|\mathbf{W}^{(T)} - \mathbf{W}^*\|_F^2}{2\eta T}$$

$$\leq \widehat{L}_S(\mathbf{W}^*, \tau) + \epsilon/4 + \epsilon/4$$

$$= \widehat{L}_S(\mathbf{W}^*, \tau) + \epsilon/2,$$

where the second inequality is by $\eta \leq \epsilon/(4L^2)$ and $T = 4\|\mathbf{W}^{(0)} - \mathbf{W}^*\|_F^2/(\eta \epsilon)$. Therefore, there exist $t' \leq T - 1$ such that $\widehat{L}_S(\mathbf{W}^{(t')}, \tau) \leq \widehat{L}_S(\mathbf{W}^*, \tau) + \epsilon/2$. Let $\widehat{T}$ to be the first time that $\widehat{L}_S(\mathbf{W}^{(\widehat{T})}, \tau) \leq \widehat{L}_S(\mathbf{W}^*, \tau) + \epsilon/2$. Again take telescope sum of (F.1) from 0 to $\widehat{T} - 1$, we have that

$$\|\mathbf{W}^{(\widehat{T})} - \mathbf{W}^*\|_F^2 \leq 2\eta \widehat{T} \widehat{L}_S(\mathbf{W}^*, \tau) - 2\eta \widehat{T} \sum_{t=0}^{\widehat{T}-1} \widehat{L}_S(\mathbf{W}^{(t)}, \tau) + 2\eta^2 L^2 \widehat{T} + \|\mathbf{W}^{(0)} - \mathbf{W}^*\|_F^2$$

$$\leq -\eta \widehat{T} \epsilon + 0.5\eta \widehat{T} \epsilon + \|\mathbf{W}^{(0)} - \mathbf{W}^*\|_F^2$$

$$\leq \|\mathbf{W}^{(0)} - \mathbf{W}^*\|_F^2,$$

where the second inequality is due to the definition of $\widehat{T}$, the last inequality is due to $-0.5\eta\widehat{T}\epsilon \leq 0$. Therefore, within $T = 4\|\mathbf{W}^{(0)} - \mathbf{W}^*\|_F^2/(\eta\epsilon)$ we can find $\widehat{\mathbf{W}} = \mathbf{W}^{(\widehat{T})}$ such that $\widehat{L}_S(\widehat{\mathbf{W}}, \tau) \leq \widehat{L}_S(\mathbf{W}^*, \tau) + \epsilon/2$ and

$$\|\mathbf{W}^{(\widehat{T})}\|_F^2 \leq 2\|\mathbf{W}^*\|_F + \|\mathbf{W}^{(0)}\|_F^2$$

where the inequality is by triangle inequality. Therefore, for any $\mathbf{x}, \mathbf{y}$

$$\begin{aligned}
\widehat{f}(\mathbf{x}, \mathbf{y}) &= \langle \mathbf{W}^*\mathbf{x}, \mathbf{y} \rangle + \langle \widehat{\mathbf{W}} - \mathbf{W}^*\mathbf{x}, \mathbf{y} \rangle \\
&\leq 1 + \|\widehat{\mathbf{W}} - \mathbf{W}^*\|_F \|\mathbf{x}\mathbf{y}^\top\|_F \\
&\leq 1 + \|\widehat{\mathbf{W}} - \mathbf{W}^*\|_F \|\mathbf{G}\|_2 \|\mathbf{H}\|_2 (R^2 + 1) \\
&\leq 1 + \|\mathbf{W}^* - \mathbf{W}^{(0)}\|_F \|\mathbf{G}\|_2 \|\mathbf{H}\|_2 (R^2 + 1).
\end{aligned}$$

Therefore the function $\widehat{f}$ is bonded by $M = 1 + \|\mathbf{W}^* - \mathbf{W}^{(0)}\|_F \|\mathbf{G}\|_2 \|\mathbf{H}\|_2 (R^2 + 1)$. Moreover, the function $\widehat{f}$ must belong to the class $\mathcal{F} = \{\langle \mathbf{W}\mathbf{x}, \mathbf{y} \rangle | \|\mathbf{W}\|_F \leq 2\|\mathbf{W}^*\|_F + \|\mathbf{W}^{(0)}\|_F^2\}$. Since the linear function class $\mathcal{F}$ has finite covering the set $\mathcal{N}(\mathcal{F}, \epsilon)$ (Bartlett & Mendelson, 2002; Zhang, 2002), by Theorem 3.3 we know that when $n \geq (8\tau^{-1}\epsilon^{-2}M \log B) \log(2\mathcal{N}(\mathcal{F}, \epsilon/32M)/\delta)$, with probability at least $1 - \delta$ we have that

$$|\widehat{L}_S(\widehat{f}, \tau) - L_{\mathcal{D}^B}(\widehat{f}, \tau)| \leq \epsilon/4$$
$$|\widehat{L}_S(f^*, \tau) - L_{\mathcal{D}^B}(f^*, \tau)| \leq \epsilon/4.$$

Thus, we can conclude that

$$\begin{aligned}
\widehat{L}_{\mathcal{D}^B}(\widehat{f}, \tau) - \widehat{L}_{\mathcal{D}^B}(f^*, \tau) &\leq \widehat{L}_S(\widehat{f}, \tau) - \widehat{L}_S(f^*, \tau) + |\widehat{L}_S(\widehat{f}, \tau) - L_{\mathcal{D}^B}(\widehat{f}, \tau)| \\
&\quad + |\widehat{L}_S(f^*, \tau) - L_{\mathcal{D}^B}(f^*, \tau)| \\
&\leq \epsilon/2 + \epsilon/4 + \epsilon/4 \\
&= \epsilon.
\end{aligned}$$

where the first inequality is by the triangle inequality, the second inequality is by the bounded gap between empirical and population loss. $\qquad\square$

*Proof of Theorem 5.6.*

$$\begin{aligned}
\mathbb{E}\Big[\|\mathbf{g}(\mathbf{x}) - \mathbf{y}\|_2^2 \Big| \mathbf{z}\Big] &= \mathbb{E}\Big[\|\mathbf{g}(\mathbf{x}) - \mathbb{E}[\mathbf{y}|\mathbf{z}] + \mathbb{E}[\mathbf{y}|\mathbf{z}] - \mathbf{y}\|_2^2 \Big| \mathbf{z}\Big] \\
&= \mathbb{E}\Big[\|\mathbf{g}(\mathbf{x}) - \mathbb{E}[\mathbf{y}|\mathbf{z}]\|_2^2 \Big| \mathbf{z}\Big] + \mathbb{E}\Big[\|\mathbb{E}[\mathbf{y}|\mathbf{z}] - \mathbf{y}\|_2^2 \Big| \mathbf{z}\Big]
\end{aligned}$$

where the second equality is due to $\mathbf{x} \perp \mathbf{y}|\mathbf{z}$ and $\mathbb{E}\Big[\mathbb{E}[y|z] - y \Big| \mathbf{z}\Big] = \mathbf{0}$. Then taking a total expectation over both sides over $\mathbf{z}$ gives that

$$\mathbb{E}\big[\|\mathbf{g}(\mathbf{x}) - \mathbf{y}\|_2^2\big] = \mathbb{E}\big[\|\mathbf{g}(\mathbf{x}) - \mathbb{E}[\mathbf{y}|\mathbf{z}]\|_2^2\big] + \mathbb{E}\big[\|\mathbf{y} - \mathbb{E}[\mathbf{y}|\mathbf{z}]\|_2^2\big] \geq \mathbb{E}\big[\|\mathbf{y} - \mathbb{E}[\mathbf{y}|\mathbf{z}]\|_2^2\big].$$

Obviously, $\mathbb{E}\big[\|\mathbf{g}(\mathbf{x}) - \mathbf{y}\|_2^2\big]$ achieves global minima when

$$\mathbf{g}(\mathbf{x}) = \mathbb{E}[\mathbf{y}|\mathbf{z}] = \mathbf{H}\begin{bmatrix} \mathbf{z} \\ \mathbb{E}[\zeta|\mathbf{z}] \end{bmatrix}.$$

This function $\mathbf{g}$ is also achievable. We can construct function $\mathbf{g}_2(\mathbf{z}) = \mathbf{H}\begin{bmatrix} \mathbf{z} \\ \mathbb{E}[\zeta|\mathbf{z}] \end{bmatrix}$, and projection function $\mathbf{g}_1(\mathbf{x}) = \mathbf{z}$ that is linear. Then we can define $\mathbf{g} = \mathbf{g}_2 \circ \mathbf{g}_1$. $\qquad\square$

*Proof of Corollary 5.7.* Since $\zeta$ is independent with $\mathbf{z}$, we have that

$$\mathbf{g}(\mathbf{x}) = \mathbf{H}\begin{bmatrix} \mathbf{z} \\ \mathbb{E}[\zeta|\mathbf{z}] \end{bmatrix} = 1/3 \cdot \begin{bmatrix} \mathbf{z} \\ \mathbf{e}_1 \\ \mathbf{0} \end{bmatrix} + 2/3 \cdot \begin{bmatrix} \mathbf{z} \\ \mathbf{e}_2 \\ \mathbf{0} \end{bmatrix}.$$

Besides, we have that

$$\mathbf{y}' = \mathbf{H} \begin{bmatrix} \mathbf{z}' \\ \zeta' \end{bmatrix} = \begin{bmatrix} \mathbf{z}' \\ \zeta' \\ \mathbf{0.} \end{bmatrix}$$

**Inner product similarity.** We have that $f(\mathbf{x}, \mathbf{y}') = \langle \mathbf{z}, \mathbf{z}' \rangle + 1/3 + 1/3 \cdot \mathbb{1}(\zeta' = \mathbf{e}_2)$. Since margin $\gamma < 1/3$. There exist $j, k$ such that $\langle \mathbf{v}_j, \mathbf{v}_k \rangle > 2/3$. Then for $\mathbf{z} = \mathbf{v}_j$, we will sample $K$ prompt $\mathbf{y}_1, \ldots, \mathbf{y}_K$. When $\mathbf{y}_j = \begin{bmatrix} \mathbf{v}_j \\ \mathbf{e}_1 \\ \mathbf{0.} \end{bmatrix}$ and $\mathbf{y}_k = \begin{bmatrix} \mathbf{v}_k \\ \mathbf{e}_2 \\ \mathbf{0.} \end{bmatrix}$, we have that

$$f(\mathbf{x}, \mathbf{y}_j) = 4/3 < \langle \mathbf{v}_j, \mathbf{v}_k \rangle + 2/3 = f(\mathbf{x}, \mathbf{y}_k),$$

which leads to the wrong top-1 prediction. The key insight behind this consequence is that $f(\mathbf{x}, \mathbf{y}') = \langle \mathbf{z}, \mathbf{z}' \rangle + 1/3 + 1/3 \cdot \mathbb{1}(\zeta' = \mathbf{e}_2)$ is greatly influenced by the unique feature $\zeta$. A similar case also exists for $\mathbf{z} = \mathbf{v}_k$ with $\mathbf{y}_j = \begin{bmatrix} \mathbf{v}_j \\ \mathbf{e}_2 \\ \mathbf{0.} \end{bmatrix}$ and $\mathbf{y}_k = \begin{bmatrix} \mathbf{v}_k \\ \mathbf{e}_1 \\ \mathbf{0.} \end{bmatrix}$. The probability that the above event occurs is at least $2/K \cdot 1/3 \cdot 2/3 = 4/(9K) \geq 1/(3K)$. Therefore, the test error is at least $1/(3K)$.

**Cosine similarity.** Notice that $\|\mathbf{g}(\mathbf{x})\|_2 = \sqrt{1 + 1/9 + 4/9} = \sqrt{14}/3$, and $\|\mathbf{y}\|_2 = 1$, therefore the cosine similarity is proportional to inner product similarity with factor $\sqrt{14}/3$. Thus, the test error is still at least $1/(3K)$.

$L_2$ **similarity.** We have that $f(\mathbf{x}, \mathbf{y}') = -\|\mathbf{z} - \mathbf{z}'\|_2^2 - 8/9 + 2/3 \cdot \mathbb{1}(\zeta' = \mathbf{e}_2)$. Since margin $\gamma < 1/3$. There exist $j, k$ such that $\|\mathbf{v}_j - \mathbf{v}_k\|_2^2 < 2/3$. Then for $\mathbf{z} = \mathbf{v}_j$, we will sample $K$ prompt $\mathbf{y}_1, \ldots, \mathbf{y}_K$. When $\mathbf{y}_j = \begin{bmatrix} \mathbf{v}_j \\ \mathbf{e}_1 \\ \mathbf{0.} \end{bmatrix}$ and $\mathbf{y}_k = \begin{bmatrix} \mathbf{v}_k \\ \mathbf{e}_2 \\ \mathbf{0.} \end{bmatrix}$, we have that

$$f(\mathbf{x}, \mathbf{y}_j) = -8/9 < -\|\mathbf{v}_j, \mathbf{v}_k\|_2^2 + 2/3 = f(\mathbf{x}, \mathbf{y}_k),$$

which leads to the wrong top-1 prediction. The key insight behind this consequence is that $f(\mathbf{x}, \mathbf{y}') = -\|\mathbf{z} - \mathbf{z}'\|_2^2 - 8/9 + 2/3 \cdot \mathbb{1}(\zeta' = \mathbf{e}_2)$ is greatly influenced by the unique feature $\zeta$. A similar case also exists for $\mathbf{z} = \mathbf{v}_k$ with $\mathbf{y}_j = \begin{bmatrix} \mathbf{v}_j \\ \mathbf{e}_2 \\ \mathbf{0.} \end{bmatrix}$ and $\mathbf{y}_k = \begin{bmatrix} \mathbf{v}_k \\ \mathbf{e}_1 \\ \mathbf{0.} \end{bmatrix}$. The probability that the above event occurs is at least $2/K \cdot 1/3 \cdot 2/3 = 4/(9K) \geq 1/(3K)$. Therefore, the test error is at least $1/(3K)$. $\qquad\square$

## G    PROOF OF RESULTS IN SECTION 6

*Proof of Corollary 6.1.* For $(\mathbf{x}, \mathbf{z}) \sim \mathcal{D}_{\mathbf{x} \times \mathbf{z}}$, $\{\mathbf{y}_k \sim \mathcal{D}_{\mathbf{y}|\mathbf{v}_k}, k \in [K]\}$, let $\mathbf{y}^* = \sum_{k \in [K]} \mathbb{1}(\mathbf{z} = \mathbf{v}_k)\mathbf{y}_k$. Denote $\mathcal{E}$ to be the event that the top-1 choice gives the wrong prediction or the margin is smaller than $\tau$. Then we have that,

$$\epsilon' \geq \mathbb{E}\left[ \log\left( \sum_{k \in [K]} \exp\left( [\widehat{f}(\mathbf{x}, \mathbf{y}_k) - \widehat{f}(\mathbf{x}, \mathbf{y}^*)]/\tau \right) \right) \right]$$

$$\geq \mathbb{E}\left[ \mathbb{1}(\mathcal{E}) \log\left( \sum_{k \in [K]} \exp\left( [\widehat{f}(\mathbf{x}, \mathbf{y}_k) - \widehat{f}(\mathbf{x}, \mathbf{y}^*)]/\tau \right) \right) \right]$$

$$\geq \mathbb{E}\left[ \mathbb{1}(\mathcal{E}) \log(1 + \exp(-1)) \right]$$

$$= \mathbb{P}(\mathcal{E}) \log(1 + e^{-1}),$$

where the first inequality is by the first bullet of Theorem 4.2, the second inequality is due to the fact that $\log\left( \sum_{k \in [K]} \exp\left( [\widehat{f}(\mathbf{x}, \mathbf{y}_k) - \widehat{f}(\mathbf{x}, \mathbf{y}^*)]/\tau \right) \right) > 0$, the last inequality is due to $\log\left( \sum_{k \in [K]} \exp\left( [\widehat{f}(\mathbf{x}, \mathbf{y}_k) - \widehat{f}(\mathbf{x}, \mathbf{y}^*)]/\tau \right) \right) \geq \log(1 + e^{-1})$ since there exists at least one similarity score $\widehat{f}(\mathbf{x}, \mathbf{y}_k)$ greater than $\widehat{f}(\mathbf{x}, \mathbf{y}^*) - \tau$ with $\mathbf{y}_k \neq \mathbf{y}^*$. Therefore, we have that $\mathbb{P}(\mathcal{E}) \leq \epsilon'/\log(1 + e^{-1}) \leq 4\epsilon'$ which completes the proof. $\qquad\square$

*Proof of Theorem 6.2.* Consider the simplest setting where $\boldsymbol{\xi}$ and $\boldsymbol{\zeta}$ are all zero vectors, and we can access to the population loss and its gradient (notice that we are constructing the negative example). We will show that even under this ideal setting, the learned score function with corresponding representations may not achieve a margin greater than $\widetilde{O}(\tau)$. Notice that

$$
\nabla_{\mathbf{W}}\mathbb{E}_{\mathcal{D}^B}L(f,\tau) = \nabla_{\mathbf{W}}\mathbb{E}\left[\log\left(\sum_{t\in[B]}\exp\left([f(\mathbf{x}_1,\mathbf{y}_t)-f(\mathbf{x}_1,\mathbf{y}_1)]/\tau\right)\right)\right]
$$

$$
+\nabla_{\mathbf{W}}\mathbb{E}\left[\log\left(\sum_{t\in[B]}\exp\left([f(\mathbf{x}_t,\mathbf{y}_1)-f(\mathbf{x}_1,\mathbf{y}_1)]/\tau\right)\right)\right]
$$

$$
=\mathbb{E}\left[\nabla_{\mathbf{W}}\log\left(\sum_{t\in[B]}\exp\left([f(\mathbf{x}_1,\mathbf{y}_t)-f(\mathbf{x}_1,\mathbf{y}_1)]/\tau\right)\right)\right]
$$

$$
+\mathbb{E}\left[\nabla_{\mathbf{W}}\log\left(\sum_{t\in[B]}\exp\left([f(\mathbf{x}_t,\mathbf{y}_1)-f(\mathbf{x}_1,\mathbf{y}_1)]/\tau\right)\right)\right]
$$

$$
=\mathbb{E}\left[\sum_{t\in[B]}\frac{\exp\left([f(\mathbf{x}_1,\mathbf{y}_t)-f(\mathbf{x}_1,\mathbf{y}_1)]/\tau\right)}{\sum_s\exp\left([f(\mathbf{x}_1,\mathbf{y}_s)-f(\mathbf{x}_1,\mathbf{y}_1)]/\tau\right)}(\mathbf{y}_t-\mathbf{y}_1)\mathbf{x}_1^\top\right]
$$

$$
+\mathbb{E}\left[\sum_{t\in[B]}\frac{\exp\left([f(\mathbf{x}_t,\mathbf{y}_1)-f(\mathbf{x}_1,\mathbf{y}_1)]/\tau\right)}{\sum_s\exp\left([f(\mathbf{x}_s,\mathbf{y}_1)-f(\mathbf{x}_1,\mathbf{y}_1)]/\tau\right)}\mathbf{y}_1(\mathbf{x}_t-\mathbf{x}_1)^\top\right]
$$

$$
=\mathbb{E}\left[\sum_{t\in[B]}\frac{\mathbb{1}(\mathbf{z}_t\neq\mathbf{z}_1)\exp\left([f(\mathbf{x}_1,\mathbf{y}_t)-f(\mathbf{x}_1,\mathbf{y}_1)]/\tau\right)}{\sum_s\exp\left([f(\mathbf{x}_1,\mathbf{y}_s)-f(\mathbf{x}_1,\mathbf{y}_1)]/\tau\right)}(\mathbf{y}_t-\mathbf{y}_1)\mathbf{x}_1^\top\right]
$$

$$
+\mathbb{E}\left[\sum_{t\in[B]}\frac{\mathbb{1}(\mathbf{z}_t\neq\mathbf{z}_1)\exp\left([f(\mathbf{x}_t,\mathbf{y}_1)-f(\mathbf{x}_1,\mathbf{y}_1)]/\tau\right)}{\sum_s\exp\left([f(\mathbf{x}_s,\mathbf{y}_1)-f(\mathbf{x}_1,\mathbf{y}_1)]/\tau\right)}\mathbf{y}_1(\mathbf{x}_t-\mathbf{x}_1)^\top\right]
$$

where the last inequality is by $\mathbf{x}_t=\mathbf{x}_1$ and $\mathbf{y}_t=\mathbf{y}_1$ when $\mathbf{z}_t=\mathbf{z}_1$. Therefore suppose function $f$ can achieve a margin greater than $\log\left(16\|\mathbf{G}\|_2^2\|\mathbf{H}\|_2^2(R^2+1)^2B\tau^{-1}\eta T\right)\tau$, we have that the gradient

$$
\left\|\nabla_{\mathbf{W}}\mathbb{E}_{\mathcal{D}^B}L(f,\tau)\right\|_F
$$

$$
\leq 2\|\mathbf{G}\|_2\|\mathbf{H}\|_2(R^2+1)\cdot\mathbb{E}\left[\sum_{t\in[B]}\frac{\mathbb{1}(\mathbf{z}_t\neq\mathbf{z}_1)\exp\left([f(\mathbf{x}_1,\mathbf{y}_t)-f(\mathbf{x}_1,\mathbf{y}_1)]/\tau\right)}{\sum_s\exp\left([f(\mathbf{x}_1,\mathbf{y}_s)-f(\mathbf{x}_1,\mathbf{y}_1)]/\tau\right)}\right]
$$

$$
+2\|\mathbf{G}\|_2\|\mathbf{H}\|_2(R^2+1)\cdot\mathbb{E}\left[\sum_{t\in[B]}\frac{\mathbb{1}(\mathbf{z}_t\neq\mathbf{z}_1)\exp\left([f(\mathbf{x}_t,\mathbf{y}_1)-f(\mathbf{x}_1,\mathbf{y}_1)]/\tau\right)}{\sum_s\exp\left([f(\mathbf{x}_s,\mathbf{y}_1)-f(\mathbf{x}_1,\mathbf{y}_1)]/\tau\right)}\right]
$$

$$
\leq 2\|\mathbf{G}\|_2\|\mathbf{H}\|_2(R^2+1)\cdot\mathbb{E}\left[\mathbb{1}(\mathbf{z}_t\neq\mathbf{z}_1)\sum_{t\in[B]}\exp\left([f(\mathbf{x}_1,\mathbf{y}_t)-f(\mathbf{x}_1,\mathbf{y}_1)]/\tau\right)\right]
$$

$$
+2\|\mathbf{G}\|_2\|\mathbf{H}\|_2(R^2+1)\cdot\mathbb{E}\left[\sum_{t\in[B]}\mathbb{1}(\mathbf{z}_t\neq\mathbf{z}_1)\exp\left([f(\mathbf{x}_t,\mathbf{y}_1)-f(\mathbf{x}_1,\mathbf{y}_1)]/\tau\right)\right]
$$

$$
\leq 0.25\tau\|\mathbf{G}\|_2^{-1}\|\mathbf{H}\|_2^{-1}(R^2+1)^{-1}\eta^{-1}T^{-1}, \tag{G.1}
$$

is very small. Now suppose the SGD trajectory start at $\mathbf{W}^{(0)} = 2\log\left(16\|\mathbf{G}\|_2^2\|\mathbf{H}\|_2^2(R^2+1)^2B\tau^{-1}\eta T\right)\cdot(\tau/\gamma)\mathbf{W}^*$. Obviously the score function with weight $\mathbf{W}^{(0)}$ achieve a margin $2\log\left(16\|\mathbf{G}\|_2^2\|\mathbf{H}\|_2^2(R^2+1)^2B\tau^{-1}\eta T\right)\tau$. Suppose there exists a time $t\leq T$ such that $\langle\mathbf{W}^{(t)}\mathbf{x},\mathbf{y}\rangle$ can achieve margin larger than $3\log\left(16\|\mathbf{G}\|_2^2\|\mathbf{H}\|_2^2(R^2+1)^2B\tau^{-1}\eta T\right)\tau$ or can achieve margin larger than $\log\left(16\|\mathbf{G}\|_2^2\|\mathbf{H}\|_2^2(R^2+1)^2B\tau^{-1}\eta T\right)\tau$. Then there must exist a first time $t<t'$ such

that the margin at time $t$ lies outsize the range between $\log\left(16\|\mathbf{G}\|_2^2\|\mathbf{H}\|_2^2(R^2+1)^2B\tau^{-1}\eta T\right)\tau$ and $3\log\left(16\|\mathbf{G}\|_2^2\|\mathbf{H}\|_2^2(R^2+1)^2B\tau^{-1}\eta T\right)\tau$. By definition of $t$ (margin gap), we know that there exist $\mathbf{x},\mathbf{y}$ such that $|\langle\mathbf{W}^{(t)}\mathbf{x},\mathbf{y}\rangle - \langle\mathbf{W}^{(0)}\mathbf{x},\mathbf{y}\rangle| > \tau$. On the other hand, we have that

$$
\begin{aligned}
\left|\langle\mathbf{W}^{(t)}\mathbf{x},\mathbf{y}\rangle - \langle\mathbf{W}^{(0)}\mathbf{x},\mathbf{y}\rangle\right| &\leq \|\mathbf{W}^{(t)}-\mathbf{W}^{(0)}\|_F\|\mathbf{x}\mathbf{y}^\top\|_F \\
&\leq 2\|\mathbf{G}\|_2\|\mathbf{H}\|_2(R^2+1)\|\mathbf{W}^{(t)}-\mathbf{W}^{(0)}\|_F \\
&\leq 2\|\mathbf{G}\|_2\|\mathbf{H}\|_2(R^2+1)\cdot\eta T\cdot 0.25\tau\|\mathbf{G}\|_2^{-1}\|\mathbf{H}\|_2^{-1}(R^2+1)^{-1}\eta^{-1}T^{-1} \\
&\leq 0.5\tau,
\end{aligned}
$$

a contradiction! Therefore, such a $t$ doesn't exist. The score function learned by SGD within $T$ iterations can't achieve a margin greater than $3\log\left(16\|\mathbf{G}\|_2^2\|\mathbf{H}\|_2^2(R^2+1)^2B\tau^{-1}\eta T\right)\tau$. $\qquad\square$

**Theorem G.1** (Formal statement of Theorem 6.3). Under the same condition as Theorem 5.5, with $\boldsymbol{\zeta}=\mathbf{0}$. (This problem setting includes the special case considered in Theorem 6.2.) Let $\epsilon \leq \lambda\gamma^2\min p_k/(3200\|\mathbf{H}\|_2^2)$ and $\tau \leq \gamma/\log(\gamma^2\min p_k/(6400B\|\mathbf{H}\|_2^2))$, within polynomial iterations, we can find a score function $\widehat{f}$ with large margin. In particular, with a probability of at least 0.99, the top-1 result gives the correct label with a margin of at least $0.5\gamma$.

*Proof.* For simplicity, consider the case that we can access the population loss and its gradient, i.e., $n \to \infty$. The regularized loss then becomes,

$$
L^{new} = L_{\mathcal{D}^B}(f,\tau) + \lambda\mathbb{E}[\|\mathbf{g}(\mathbf{x})-\mathbf{h}(\mathbf{y})\|_2^2].
$$

Since the new loss is still convex and even strongly convex. By applying the same technique in the proof of the Theorem 5.5, within polynomial iterations, we can find $L^{new}(f,\tau,\lambda) \leq L^{new}(f^*,\tau,\lambda)+\epsilon$. Besides,

$$
L^{new}(f^*,\tau,\lambda) = L_{\mathcal{D}^B}(f^*,\tau) \leq 2\mathbb{E}\left[\log\left(\sum_{t\in[B]}\mathbb{1}(\mathbf{z}_t=\mathbf{z}_1)\right)\right] + 2B\exp(-\gamma/\tau)
$$

where the first equality is by plugging in $\mathbf{W}^* = \mathbf{H}(\mathbf{H}^\top\mathbf{H})^{-1}\mathbf{P}(\mathbf{G}^\top\mathbf{G})^{-1}\mathbf{G}^\top$, $\mathbf{g}(\mathbf{x})=\mathbf{W}\mathbf{x}$, $\mathbf{h}(\mathbf{y})=\mathbf{y}$, the inequality is by Lemma E.3. Thus we have that

$$
L_{\mathcal{D}^B}(f,\tau) + \lambda\mathbb{E}[\|\mathbf{g}(\mathbf{x})-\mathbf{h}(\mathbf{y})\|_2^2] \leq 2\mathbb{E}\left[\log\left(\sum_{t\in[B]}\mathbb{1}(\mathbf{z}_t=\mathbf{z}_1)\right)\right] + \epsilon',
$$

where $\epsilon' = \epsilon + 2B\exp(-\gamma/\tau)$. By (E.7) and (E.8), we know that $L_{\mathcal{D}^B}(f,\tau) \geq 2\mathbb{E}\left[\log\left(\sum_{t\in[B]}\mathbb{1}(\mathbf{z}_t=\mathbf{z}_1)\right)\right]$. Therefore, we can conclude that

$$
\mathbb{E}[\|\mathbf{g}(\mathbf{x})-\mathbf{h}(\mathbf{y})\|_2^2] \leq \epsilon'/\lambda \leq \gamma^2\min p_k/(1600\|\mathbf{H}\|_2^2),
$$

where the last inequality is by choose $\epsilon \leq \lambda\gamma^2\min p_k/(3200\|\mathbf{H}\|_2^2)$ and $\tau \leq \gamma/\log(\gamma^2\min p_k/(6400B\|\mathbf{H}\|_2^2))$. Then by Chebyshev's inequality, for any $\mathbf{z}$, with probability $1-0.01$ we have $\|\mathbf{g}(\mathbf{x})-\mathbf{h}(\mathbf{y})\|_2 \leq \sqrt{100\max p_k^{-1}\mathbb{E}[\|\mathbf{g}(\mathbf{x})-\mathbf{h}(\mathbf{y})\|_2^2]} \leq \gamma/(4\|\mathbf{H}\|_2)$. Then for any $\mathbf{y}'$ that has the different shared feature from $\mathbf{y}$ (i.e., $\mathbf{z}' \neq \mathbf{z}$) we have that

$$
\begin{aligned}
&\langle\mathbf{g}(\mathbf{x}),\mathbf{h}(\mathbf{y}')\rangle - \langle\mathbf{g}(\mathbf{x}),\mathbf{h}(\mathbf{y})\rangle \\
&\leq \langle\mathbf{h}(\mathbf{y}),\mathbf{h}(\mathbf{y}')\rangle - \langle\mathbf{h}(\mathbf{y}),\mathbf{h}(\mathbf{y})\rangle + \|\mathbf{g}(\mathbf{x})-\mathbf{h}(\mathbf{y})\|_2\cdot(\|\mathbf{h}(\mathbf{y}')\|_2+\|\mathbf{h}(\mathbf{y})\|_2) \\
&\leq -\gamma+\gamma/2 \\
&\leq -\gamma/2,
\end{aligned}
$$

where the first inequality is by triangle inequality, the second inequality is by $\|\mathbf{g}(\mathbf{x})-\mathbf{h}(\mathbf{y})\|_2 \leq \gamma/(4\|\mathbf{H}\|_2)$ and $\|\mathbf{h}(\mathbf{y}')\|_2 = \|\mathbf{h}(\mathbf{y})\|_2 \leq \|\mathbf{H}\|_2$ since $\boldsymbol{\zeta}=\mathbf{0}$. $\qquad\square$

