# OpenReview forum: "Understanding Transferable Representation Learning and Zero-shot Transfer in CLIP"
_ICLR.cc/2024/Conference — ICLR 2024 poster_

### Official Review · Reviewer_nFDE · 2023-10-31

**Soundness:** 3 good
**Presentation:** 3 good
**Contribution:** 2 fair
**Rating:** 6
**Confidence:** 3

**Summary:**

This paper theoretically examines transferable representation learning of CLIP. The analysis reveals that with a near-optimal network trained on the data, features from different modalities align, allowing for zero-shot learning when appropriate prompts are used. The paper also demonstrates that contrastive learning with sparse features can lead to unexpected positive pairs, emphasizing the need for careful consideration. Building on these general theoretical findings, the authors provide deeper insights into specific cases, illustrating how multi-modal learning aligns different features and how CLIP's learned features outperform those obtained through naive square loss. To validate their theoretical predictions, the authors conduct experiments on real data. Additionally, inspired by their theoretical findings, they propose a novel regularization technique for CLIP, effectively improving zero-shot performance across various tasks, as confirmed by empirical results.

**Strengths:**

1. The paper is well-written and flows smoothly, making it relatively easy for readers to understand.
2. This article focuses on what's behind the explosive effectiveness of CLIP, and the paper attempts to delve into the principles underlying CLIP, demonstrating a certain level of originality and innovation.

**Weaknesses:**

1. More relevant experimental results are expected, such as results from a wider range of downstream tasks.
2. The article exclusively analyzes and experiments with CLIP, without thoroughly exploring the applicability of this new methods to other relevant contrastive learning approaches.

**Questions:**

Can one simply replace the CLIP component in all CLIP-related work with CLIP+Reg to achieve a better performance?

---

> ### Author Response · Authors · 2023-11-19
> **Response to Reviewer nFDE**
>
> Thank you for your helpful comments!
>
> **Q1**: More relevant experimental results are expected, such as results from a wider range of downstream tasks.
>
> **A1**: Thank you for your suggestion. To add more experiments on downstream tasks, we have added experiments on retrieval accordingly and summarized the zero-shot results in table below, including Flickr-30k (1k) and MSCOCO (5k) following the CyCLIP paper [1]. As also noted in [1], text retrieval per image is easier than image retrieval per caption, due to the nature of both datasets containing 5 captions for each image. The trend of our results align with that reported in [1]. Lastly, while CyCLIP does not provide consistent improvement over CLIP (in our experiment setting as well as in their reported results), our regularization term provides consistent improvement to both CLIP objective and CyCLIP objective. This has also been updated in our Appendix C.1.
>
> Flickr-30k (1k)
> |  | Text R@1 | Text R@5 | Text R@10 | Image R@1 | Image R@5 | Image R@10 | Average
> | :--------: | :-------: |  :--------: | :-------: | :-------: | :-------: | :-------: | :-------: |
> | CLIP | 87.36 | 93.0 | 95.18 | 26.88 | 54.18 | 66.22 | 70.47 |
> | CLIP+Reg | **87.42** | **93.42** | **95.82** | **29.94** | **58.00** | **69.82** | **72.40** |
> | CyCLIP | 87.34 | 93.12 | 95.04 | 29.00 | 56.50 | 67.62 | 71.44 |
> | CyCLIP+Reg | 87.20 | 93.20 | 95.56 | 29.14 | 56.94 | 68.64 | 71.78 |
>
> MSCOCO (5k)
> |  | Text R@1 | Text R@5 | Text R@10 | Image R@1 | Image R@5 | Image R@10 | Average
> | :--------: | :-------: |  :--------: | :-------: | :-------: | :-------: | :-------: | :-------: |
> | CLIP | 81.19 | 83.21 | 84.42 | 4.73 | 11.66 | 15.93 | 46.86 |
> | CLIP+Reg | **81.25** | **83.31** | 84.49 | 4.98 | 12.14 | 16.66 | 47.14 |
> | CyCLIP | 81.06 | 82.92 | 84.28 | 4.70 | 11.66 | 15.93 | 46.86 |
> | CyCLIP+Reg | 81.31 | 83.28 | **84.65** | **5.27** | **12.17** | **16.70** |**47.23** |
>
> ---
>
> **Q2**: The article exclusively analyzes and experiments with CLIP, without thoroughly exploring the applicability of this new methods to other relevant contrastive learning approaches.
>
> **A2**: Since we are the first paper that theoretically studies the transferable representation learning of CLIP,  the primary focus of our paper was an in-depth analysis of the CLIP model. Due to the space limit, we only consider the vanilla CLIP model in our theory and, additionally, the CyCLIP [1] model in the experiment. We acknowledge the importance of exploring other contrastive learning approaches and consider this an important direction for future research.
>
> ---
>
> **Q3**: Can one simply replace the CLIP component in all CLIP-related work with CLIP+Reg to achieve a better performance?
>
> **A3**: We think so. Replacing the CLIP component with CLIP+Reg in various CLIP-related projects is feasible and can potentially lead to improved performance. In principle, our regularization is quite simple and can improve the alignment between image-text pairs, as demonstrated by the above experiment results. However, achieving a successful replacement might necessitate additional effort in exploring various related works to fully leverage the benefits of our regularization.
>
> ---
>
> [1] Goel et al. "Cyclip: Cyclic contrastive language-image pretraining." Advances in Neural Information Processing Systems 35 (2022): 6704-6719.

---

### Official Review · Reviewer_CWhP · 2023-10-31

**Soundness:** 2 fair
**Presentation:** 2 fair
**Contribution:** 2 fair
**Rating:** 6
**Confidence:** 2

**Summary:**

This paper offers theoretical analysis of the underlying principles of CLIP, shedding light on why CLIP exhibits robust transferability. Additionally, the paper introduces a novel regularization technique designed to enhance the performance of CLIP.

**Strengths:**

This paper is well-written and novel. This paper offers a robust analysis, including mathematical proofs, of CLIP. These contributions greatly contribute to our understanding of CLIP.

**Weaknesses:**

1. The paper's primary focus appears to be on CLIP's zero-shot transferability. While this is undoubtedly a significant aspect, it's worth considering that CLIP's robust zero-shot performance results from a strong semantic space based on extensive vision-semantic data. Therefore, an exploration of the visual-semantic alignment aspect could be an intriguing avenue for further investigation.

2. In introduction section, the author cites "blue sky" and "white cloud" as examples of unique features. However, these instances might be seen as special cases.  As CLIP is based on a large amount of vision-semantic data, it's possible that the missing elements could appear in various other captions. Therefore, I question the significance of this problem. To address this concern, the author may need to conduct overall statistics on the data. Furthermore, the term 'unique features' could benefit from a more precise definition or explanation.

3.  Some notations and definitions in the paper can be challenging to follow. For instance, the terms 'one-to-one mapping' or 'one-to-one matching' could benefit from clearer explanations for readers.

4. Expanding the range of experiments to include various downstream tasks, rather than solely focusing on zero-shot and Linear probing, would provide a more comprehensive assessment of the paper's proposed methods and their practical applications.

**Questions:**

See `Weakness' above.

---

> ### Author Response · Authors · 2023-11-19
> **Response to Reviewer CWhP (1/2)**
>
> Thank you for your helpful and detailed comments!
>
> **Q1**: The paper's primary focus appears to be on CLIP's zero-shot transferability. While this is undoubtedly a significant aspect, it's worth considering that CLIP's robust zero-shot performance results from a strong semantic space based on extensive vision-semantic data. Therefore, an exploration of the visual-semantic alignment aspect could be an intriguing avenue for further investigation.
>
> **A1**: Thank you for your valuable suggestion.  While our primary focus is CLIP’s zero-shot transferability, we indeed considered the visual-semantic alignment in our analysis. As noted in Remark 4.3, image-text pairs sharing the same topic tend to score higher, indicating a visual-semantic alignment in CLIP's learned representations. Furthermore, the margin discussed in our paper can be considered a useful metric for studying this alignment. We've expanded on this in Appendix A for further clarification.
>
> ---
>
> **Q2**: In introduction section, the author cites "blue sky" and "white cloud" as examples of unique features. However, these instances might be seen as special cases. As CLIP is based on a large amount of vision-semantic data, it's possible that the missing elements could appear in various other captions. Therefore, I question the significance of this problem.
>
> **A2**: In the introduction section, we presented an example to illustrate the existence of such an issue. While CLIP is trained on extensive image-caption datasets, each image is paired with only a single, brief caption. It is crucial to note that CLIP's pre-training data are web-sourced image-caption pairs, characterized by their succinct and often non-elaborate nature, as these captions are not meticulously crafted by annotators. Consequently, instances where captions do not comprehensively describe the corresponding image are common in such datasets.
>
> While the ground truth object list is not available for CC3M, we can first consider the MSCOCO dataset that has a ground truth object list. Analyzing the 410,600 image-caption pairs from its training data, we identified 330,843 pairs wherein the caption failed to include at least one object based on exact matching criteria. This amounts to **80.58%** of the dataset. We have included the distribution bar plot in Appendix C.3.
>
> We also provide several examples of image captions, along with the ground truth objects in the image.  We highlight the objects that are missed from the caption.
> - Caption: “A restaurant has modern wooden tables and chairs.”
> - Objects: [**potted plant**, dining table, **book**, **vase**, chair]
>
> - Caption: “A bicycle parked in a kitchen with a stove and cabinets.”
> - Objects: [bicycle, **bottle**, **cup**, **toaster**, **sink**, **spoon**, **bowl**, **oven**]
>
> - Caption: “A person is cutting a roast with a fork and knife.”
> - Objects: [person, **oven**, knife]
>
> As shown above, humans do not extraneously identify all conceivable objects in an image when trying to describe it, let alone the background information such as the scene. If the reviewer browses through some examples of MSCOCO and CC3M, it will be a frequent case that the captions cannot fully depict the image details. In Appendix C.3, we further provide several image-caption examples from CC3M.

---

> ### Author Response · Authors · 2023-11-19
> **Response to Reviewer CWhP (2/2)**
>
> **Q3**: Some notations and definitions in the paper can be challenging to follow. ('one-to-one mapping' or 'one-to-one matching')
>
> **A3**: Thank you for the valuable suggestion, we have unified the terminology as one-to-one mapping.
>
> ---
>
> **Q4**: Expanding the range of experiments to include various downstream tasks, rather than solely focusing on zero-shot and Linear probing…
>
> **A4**: Our research concentrates on the zero-shot transfer capabilities in the CLIP. This focus aligns with the core tasks (zero-shot and linear probing) emphasized in the original CLIP paper and its subsequent developments, such as FILIP [1] and CyCLIP [4]. To add more experiments on downstream tasks, we have added experiments on retrieval accordingly and summarized the zero-shot results in table below, including Flickr-30k (1k) and MSCOCO (5k) following [4]. As also noted in [4], text retrieval per image is easier than image retrieval per caption, due to the nature of both datasets containing 5 captions for each image. The trend of our results align with that reported in [4]. Lastly, while CyCLIP does not provide consistent improvement over CLIP (in our experiment setting as well as in their reported results), our regularization term provides consistent improvement to both CLIP objective and CyCLIP objective. This has also been updated in our Appendix C.1.
>
> Flickr-30k (1k)
> |  | Text R@1 | Text R@5 | Text R@10 | Image R@1 | Image R@5 | Image R@10 | Average
> | :--------: | :-------: |  :--------: | :-------: | :-------: | :-------: | :-------: | :-------: |
> | CLIP | 87.36 | 93.0 | 95.18 | 26.88 | 54.18 | 66.22 | 70.47 |
> | CLIP+Reg | **87.42** | **93.42** | **95.82** | **29.94** | **58.00** | **69.82** | **72.40** |
> | CyCLIP | 87.34 | 93.12 | 95.04 | 29.00 | 56.50 | 67.62 | 71.44 |
> | CyCLIP+Reg | 87.20 | 93.20 | 95.56 | 29.14 | 56.94 | 68.64 | 71.78 |
>
> MSCOCO (5k)
> |  | Text R@1 | Text R@5 | Text R@10 | Image R@1 | Image R@5 | Image R@10 | Average
> | :--------: | :-------: |  :--------: | :-------: | :-------: | :-------: | :-------: | :-------: |
> | CLIP | 81.19 | 83.21 | 84.42 | 4.73 | 11.66 | 15.93 | 46.86 |
> | CLIP+Reg | **81.25** | **83.31** | 84.49 | 4.98 | 12.14 | 16.66 | 47.14 |
> | CyCLIP | 81.06 | 82.92 | 84.28 | 4.70 | 11.66 | 15.93 | 46.86 |
> | CyCLIP+Reg | 81.31 | 83.28 | **84.65** | **5.27** | **12.17** | **16.70** |**47.23** |
>
> ---
>
> [1] Yao, Lewei, et al. "FILIP: Fine-grained Interactive Language-Image Pretraining." International Conference on Learning Representations. 2021.
>
> [2] Li et al. "Supervision Exists Everywhere: A Data Efficient Contrastive Language-Image Pretraining Paradigm." International Conference on Learning Representations. 2021.
>
> [3] Mu et al. "Slip: Self-supervision meets language-image pre-training." European Conference on Computer Vision. Cham: Springer Nature Switzerland, 2022.
>
> [4] Goel et al. "Cyclip: Cyclic contrastive language-image pretraining." Advances in Neural Information Processing Systems 35 (2022): 6704-6719.

---

> ### Author Response · Authors · 2023-11-21
> **Follow up with Reviewer CWhP**
>
> Dear Reviewer CWhP,
>
> Thank you again for your insightful comments. We are following up to engage in further discussion and address any outstanding questions you might still have. If our rebuttal and the revisions to our paper have satisfactorily addressed your concerns, we respectfully hope that you could reconsider your assessment of our work.

---

### Official Review · Reviewer_7qr5 · 2023-10-31

**Soundness:** 3 good
**Presentation:** 2 fair
**Contribution:** 2 fair
**Rating:** 6
**Confidence:** 3

**Summary:**

This paper focuses on providing theoretical support for the CLIP training and its zero-shot transferability. The main claim is that the contrastive learning objective in CLIP may not cover all the positive pairs, e.g., some features in an image may not be present in its corresponding captions. In section 3, the authors show that the empirical loss converges to the true loss when the number of batches is large enough. In section 4, they show that the learned similarity score f_hat between negative pairs is smaller than the score between positive pairs given that there exists a score function f* such that this relation holds. Based on such assumption, in section 5, they conclude that a trained CLIP model can achieve small top-r error and this generalizes to different distributions as the distribution shift is bounded. Based on the prior assumption and derivations, they have three claims: 1) Margin depends on the temperature tau, 2) We should only regularize positive pairs instead of both positive and negative pairs, 3) With sufficiently small tau, we can find a f_hat with large margin. They test these claims with experiments on CC3M.

**Strengths:**

The paper provides theoretical bounds on CLIP training and its zero-shot transferability.

**Weaknesses:**

1. Contrastive learning has been widely studied in the community with several variants of NCE loss, with different ways to regularize positive and negative pairs to improve the margins (e.g., [a]). The behavior of temperature in contrastive learning was also studied (e.g., [b]), and so was regularization (e.g., [c]). It is not surprising that adding the regularization of the distance between positive pairs can improve the performance. Also, how does the proposed solution compare to those methods?

2. The authors propose only to regularize the distance between positive pairs, but there is no ablation comparison to the variants that regularize both or negative pairs only.

3. The introduction states that the contrastive learning objective in CLIP may not cover all the positive pairs, which makes sense. However,
 it is unclear how the proposed solution addresses this issue.

4. The experiments are conducted on a relatively small dataset compared to CLIP. The training behavior and the generalization ability of the representation may be different.

5. The derivations make sense but they are under several assumptions.

[a] Unified Contrastive Learning in Image-Text-Label Space, CVPR'22
[b] Understanding the Behaviour of Contrastive Loss, CVPR'21
[c] Large-Margin Contrastive Learning with Distance Polarization Regularizer, ICML’21

**Questions:**

My questions are listed in the weakness section.

---

> ### Author Response · Authors · 2023-11-19
> **Response to Reviewer 7qr5 (1/2)**
>
> Thank you for your constructive comments!
>
> **Q1**: Contrastive learning has been widely studied in the community. Also, how does the proposed solution compare to those methods?
>
> **A1**:  Thank you for pointing out these related works. We have added discussions on those papers. Similar to CLIP and ALIGN, paper (a) proposed a contrastive learning scheme UniCL for multi-modal learning. Unlike CLIP that primarily rely on image-text data, UniCL uniquely integrates image-label data (ImageNet-1K). This integration enables them to group data with identical labels, allowing for a broader range of positive pair identifications within the dataset. There is no regularization term proposed in this paper; rather, it’s a different contrastive learning method. We have included this paper in discussion as an improvement over CLIP, alongside other methods we discussed such as SLIP, FILIP, and CyCLIP. Moreover, for the paper that actually proposed regularization terms for CLIP (CyCLIP), we indeed made a thorough discussion.
>
> The (b,c) focuses on **unimodal contrastive learning**, and thus cannot adequately explain the zero-shot transfer capability of CLIP. As we emphasized in our paper, we focus on **multi-modal contrastive learning**, which has not been studied from a theoretical perspective and has its special challenges resulting from multi-modality. Ours is the first paper to provide a formal understanding of transferable representation learning and zero-shot transfer in CLIP. We have added (b,c) to the related work section and as well as the detailed discussion in Appendix A.
>
> ---
>
> **Q2**: The authors propose only to regularize the distance between positive pairs, but there is no ablation comparison to the variants that regularize both or negative pairs only.
>
> **A2**: In our paper, we did clarify that accurately identifying truly negative pairs in image-text data is challenging, as different pairs within the same batch may share common features. Since web-scale image-text data are inherently unlabeled, it is different from the unimodal case where negative pairs can be easily determined by the class labels. As shown in Figure 1, the image-text pairs containing similar features could appear in the same training batch and be mistakenly considered as negative pairs. Consequently, our proposed regularization term only incorporates positive pairs.
>
> We have added an ablation study by taking all off-diagonal pairs as negative pairs and adding them as a regularization term. The table presented below compares the effects of regularizing positive pairs (Pos) against off-diagonal pairs (Neg) in terms of zero-shot accuracy. While the regularization on positive pairs markedly improves performance, regularization on off-diagonal pairs yields only marginal enhancements on some datasets and no improvement on others. This unstable performance may be attributed to the presence of positive pairs among the off-diagonal elements in the unlabeled image-text data. We have added a discussion on this in Appendix C.2.
>
> |  | CIFAR10 | CIFAR100 | STL10 | Food101 | ImageNetV2 |  DTD | Average |
> | :--------: | :-------: |  :--------: | :-------: | :-------: | :-------: | :-------: | :-------: |
> | CLIP | 63.85 | 31.17 | 90.35 | 8.39 | 20.24 | **21.22** |  39.20 |
> | CLIP+Pos | **67.47** | **33.33** | **92.64** | **12.14** | **22.36** | 19.63 | **41.26** |
> | CLIP+Neg | 64.36 | 31.01 | 91.25 | 9.59 | 20.17 | 20.74 | 39.52 |
>
> ---
>
> **Q3**: The introduction states that the contrastive learning objective in CLIP may not cover all the positive pairs, which makes sense. However, it is unclear how the proposed solution addresses this issue.
>
> **A3**: In our paper, we acknowledge the challenge posed by the fact that "Contrastive learning objectives in CLIP may not cover all positive pairs," both from theoretical and empirical standpoints. Our primary goal is to rigorously explore transferable representation learning in CLIP, and to introduce a theoretically-guided regularization that enhances the original CLIP, particularly focusing on the concept of “margin”. Therefore, our approach primarily addresses this issue from a theoretical angle by incorporating this mismatch into our analysis. We recognize the significance of empirically addressing this issue, but it goes beyond the scope of our current work. The literature [a] you mentioned is a milestone work in this area, which introduces a new loss function. Analyzing why this improved loss [a] outperforms CLIP from a theoretical perspective can be an interesting future direction.

---

> ### Author Response · Authors · 2023-11-19
> **Response to Reviewer 7qr5 (2/2)**
>
> **Q4**: The experiments are conducted on a relatively small dataset compared to CLIP.
>
> **A4**: Unfortunately, the dataset used by Open AI is not open-source. Therefore we utilize the CC3M dataset which is a standard and widely used dataset in prior research on CLIP [1-4]. This dataset is also used in UniCL as the reviewer mentioned, referred to as GCC3M. Many of their experiments (ImageNet-1K, GCC3M, GCC3M+ImageNet-1K) are done at smaller or the same scale.
>
> ---
>
> **Q5**: The derivations make sense but they are under several assumptions.
>
> **A5**: In order to theoretically analyze CLIP, we have to make certain assumptions, like all theory papers do. The assumptions streamline the analysis, rendering the problem more amenable. More importantly, the assumptions made in our paper are quite standard and reasonable. In detail, in Assumption 3.1, we make a conditionally independent assumption which is commonly used in the theoretical literature of self-supervised learning (as we mentioned in Remark 3.2). In Assumption 4.1, we make a completeness assumption to allow the existence of a good representation of image-text pairs. We acknowledge that the relaxation of our assumptions would be a interesting future direction.
>
> [1] Yao, Lewei, et al. "FILIP: Fine-grained Interactive Language-Image Pretraining." International Conference on Learning Representations. 2021.
>
> [2] Li et al. "Supervision Exists Everywhere: A Data Efficient Contrastive Language-Image Pretraining Paradigm." International Conference on Learning Representations. 2021.
>
> [3] Mu et al. "Slip: Self-supervision meets language-image pre-training." European Conference on Computer Vision. Cham: Springer Nature Switzerland, 2022.
>
> [4] Goel et al. "Cyclip: Cyclic contrastive language-image pretraining." Advances in Neural Information Processing Systems 35 (2022): 6704-6719.

---

> ### Author Response · Authors · 2023-11-21
> **Follow up with Reviewer 7qr5**
>
> Dear Reviewer 7qr5,
>
>
> Thank you for taking the time to review our paper and providing valuable feedback. We would like to follow up with you and provide additional experiment results regarding Q2 on regularization of negative pairs. In our previous rebuttal, we have explored regularizing the negative (off-diagonal) pairs as compared to positive pairs. Here, we have provided additional experiment results on regularizing both positive and negative (off-diagonal) pairs, which are summarized in the table below. These results demonstrate that regularization of both pairs leads to improvements over only regularizing negative (off-diagonal) pairs across most datasets. However, it still underperforms our proposed method CLIP+Pos, which only regularizes positive pairs. This experiment further supports our initial observation that some off-diagonal image-text pairs may share common features, making their categorization as negative pairs potentially detrimental. Identifying true negative pairs in unlabeled image-text data could be an interesting direction for future research.
>
>
> |  | CIFAR10 | CIFAR100 | STL10 | Food101 | ImageNetV2 |  DTD | Average |
> | :--------: | :-------: |  :--------: | :-------: | :-------: | :-------: | :-------: | :-------: |
> | CLIP | 63.85 | 31.17 | 90.35 | 8.39 | 20.24 | **21.22** |  39.20 |
> | CLIP+Pos | **67.47** | **33.33** | **92.64** | **12.14** | **22.36** | 19.63 | **41.26** |
> | CLIP+Neg | 64.36 | 31.01 | 91.25 | 9.59 | 20.17 | 20.74 | 39.52 |
> | CLIP+Both | 65.12 | 32.63 | 91.68 | 9.67 | 21.07 | 20.26 | 39.91 |
>
>
> We look forward to your further feedback and are more than happy to discuss any remaining questions you may have. If our rebuttal and the revised paper have effectively addressed your questions and concerns, we sincerely hope you could reconsider your evaluation of our paper. Thank you!

---

### Official Review · Reviewer_P1Cv · 2023-11-01

**Soundness:** 3 good
**Presentation:** 3 good
**Contribution:** 3 good
**Rating:** 8
**Confidence:** 3

**Summary:**

This paper investigates transferable representation learning underlying CLIP and demonstrates how features from different modalities can be aligned. Then a new CLIP-type method is proposed, the effectiveness of the proposed method is proved through experiments on multiple benchmark datasets.

**Strengths:**

- This paper is well-written and easy to follow.
- This paper theoretically examines the transferable representation learning in CLIP. The theory seems sound.
- This paper proposes an easy regularization technique for CLIP that can effectively improve its zero-shot performance.

**Weaknesses:**

My major concerns lie in the empirical studies.
- The current pre-training experiments are all based on the CC3M, which is much smaller than the full 400M dataset used by the CLIP. It is unclear whether the proposed regularization technique holds when extended to a larger dataset. It is recommended to conduct experiments on datasets with different sizes.
- In Table 1, why incorporating the regularization term into the contrastive objective is harmful to DTD?
- It seems that the results of CyCLIP in Table 1 and Table 2 are inconsistent with the original CyCLIP paper.
- Can the proposed regularization technology work in CyCLIP?
- Is the method equally effective for other downstream tasks such as retrieval?

**Questions:**

Please see weaknesses.

---

> ### Author Response · Authors · 2023-11-19
> **Response to Reviewer P1Cv (1/2)**
>
> Thank you for your strong support and valuable feedback！We answer your major comments and questions as follows.
>
> **Q1**:  Current pre-training experiments are all based on the CC3M… it is recommended to conduct experiments on datasets with different sizes.
>
> **A1**: Thank you for your valuable suggestion. CLIP serves as a strategy for learning features that can be transferred between text and images. Unfortunately, the dataset used by Open AI is not open-source. Therefore we utilize the CC3M dataset which is a standard and widely used dataset used in prior research on CLIP [1-4].
>
> ---
>
> **Q2**: In Table 1, why incorporating the regularization term into the contrastive objective is harmful to DTD?
>
> **A2**: Thank you for raising this point. While our method can improve on 5 out of 6 downstream tasks, Describable Textures Dataset (DTD) presents a special case where the regularization term can get a worse zero shot performance. We conjecture that the unique properties of texture data in DTD require a more fine-grained approach of regularization. Therefore, we conduct more experiments on the DTD dataset and find that the combination of our regularization and CyCLIP can improve the original CLIP on the DTD data set.
>
> | CLIP | CLIP+Reg | CyCLIP | CyCLIP+Reg |
> | :--------: | :-------: |  :--------: | :-------: |
> | 21.22 | 19.63 | 20.21 | **21.49** |
>
> ---
>
> **Q3**: It seems that the results of CyCLIP in Table 1 and Table 2 are inconsistent with the original CyCLIP paper.
>
> **A3**: Indeed, our results for CyCLIP are better than the results reported in the original CyCLIP paper on most tasks. This is due to the different experiment setting of our work, as we train from pre-trained image and text encoders while they trained from scratch. Similar settings can be found in LiT [5] and Flamingo [6] that train from pre-trained single-modality models.
>
> ---
>
> **Q4**: Can the proposed regularization technology work in CyCLIP?
>
> **A4**: Yes, it indeed can be used for CyCLIP and leads to improvements on most tasks. We have added the results for CyCLIP+Reg as follows.
>
> Zero-shot
> |  | CIFAR10 | CIFAR100 | STL10 | Food101 | ImageNetV2 |  DTD | Average |
> | :--------: | :-------: |  :--------: | :-------: | :-------: | :-------: | :-------: | :-------: |
> | CyCLIP | 60.71 | 28.87 | 89.98 | 9.72 | 19.66 | 20.21 | 38.19 |
> | CyCLIP+Reg | 58.73 | **30.15** | **90.79** | **10.87** | **19.68** | **21.49** | **38.62** |
>
> Linear probing
> |  | CIFAR10 | CIFAR100 | STL10 | Food101 | DTD |  Flowers | OxfordPets | Average |
> | :--------: | :-------: |  :--------: | :-------: | :-------: | :-------: | :-------: | :-------: | :-------: |
> | CyCLIP | 86.31 | 63.93 | 93.69 | 61.57 | 56.86 | 70.56 | 70.46 | 71.91|
> | CyCLIP+Reg | 85.08 | 62.60 | **93.83** | **62.40** | 54.95 | **72.32** | **77.24** | **72.63** |
>
> ---
>
> **Q5**: Is the method equally effective for other downstream tasks such as retrieval?
>
> **A5**: Thank you for pointing us to the retrieval task. We have added experiments on retrieval accordingly and summarized the zero-shot results in the table below, including Flickr-30k (1k) and MSCOCO (5k) following the original CyCLIP paper [4]. As also noted in [4], text retrieval per image is easier than image retrieval per caption, due to the nature of both datasets containing 5 captions for each image. The trend of our results align with that reported in [4]. Lastly, while CyCLIP does not provide consistent improvement over CLIP (in our experiment as well as in their reported results), our regularization term provides consistent improvement to both CLIP objective and CyCLIP objective. This has also been updated in our Appendix C.1.
>
> Flickr-30k (1k)
> |  | Text R@1 | Text R@5 | Text R@10 | Image R@1 | Image R@5 | Image R@10 | Average
> | :--------: | :-------: |  :--------: | :-------: | :-------: | :-------: | :-------: | :-------: |
> | CLIP | 87.36 | 93.0 | 95.18 | 26.88 | 54.18 | 66.22 | 70.47 |
> | CLIP+Reg | **87.42** | **93.42** | **95.82** | **29.94** | **58.00** | **69.82** | **72.40** |
> | CyCLIP | 87.34 | 93.12 | 95.04 | 29.00 | 56.50 | 67.62 | 71.44 |
> | CyCLIP+Reg | 87.20 | 93.20 | 95.56 | 29.14 | 56.94 | 68.64 | 71.78 |
>
> MSCOCO (5k)
> |  | Text R@1 | Text R@5 | Text R@10 | Image R@1 | Image R@5 | Image R@10 | Average
> | :--------: | :-------: |  :--------: | :-------: | :-------: | :-------: | :-------: | :-------: |
> | CLIP | 81.19 | 83.21 | 84.42 | 4.73 | 11.66 | 15.93 | 46.86 |
> | CLIP+Reg | **81.25** | **83.31** | 84.49 | 4.98 | 12.14 | 16.66 | 47.14 |
> | CyCLIP | 81.06 | 82.92 | 84.28 | 4.70 | 11.66 | 15.93 | 46.86 |
> | CyCLIP+Reg | 81.31 | 83.28 | **84.65** | **5.27** | **12.17** | **16.70** |**47.23** |

---

> ### Author Response · Authors · 2023-11-19
> **Response to Reviewer P1Cv (2/2)**
>
> [1] Yao, Lewei, et al. "FILIP: Fine-grained Interactive Language-Image Pretraining." International Conference on Learning Representations. 2021.
>
> [2] Li et al. "Supervision Exists Everywhere: A Data Efficient Contrastive Language-Image Pretraining Paradigm." International Conference on Learning Representations. 2021.
>
> [3] Mu et al. "Slip: Self-supervision meets language-image pre-training." European Conference on Computer Vision. Cham: Springer Nature Switzerland, 2022.
>
> [4] Goel et al. "Cyclip: Cyclic contrastive language-image pretraining." Advances in Neural Information Processing Systems 35 (2022): 6704-6719.
>
> [5] Xiaohua Zhai et al. "Lit: Zero-shot transfer with locked-image text tuning." Proceedings of the IEEE/CVF Conference on Computer Vision and Pattern Recognition. 2022.
>
> [6] Jean-Baptiste Alayrac et al. "Flamingo: a visual language model for few-shot learning." Advances in Neural Information Processing Systems 35 (2022): 23716-23736.

---

### Meta-Review · Area_Chair_w4vs · 2023-12-06

**Metareview:**

This paper tackles an important topic: building theoretical foundations for multimodal models that can be used for zero-shot tasks. CLIP is the particular model of interest. Results including a generalization bound and a result on the learned representations that produces insights into desirable temperature parameters and batch size. The authors also study zero-shot transfer error, the impact of a non-contrastive loss (in this case the square loss), and finally propose an improved regularization method.

The strengths here are clear: a fairly intensive theoretical grounding that has many interesting implications. There are no great weaknesses; as usual in these types of works, there are important question around how realistic the assumptions are, etc.

All reviewers generally agreed about the papers' strengths. The paper is above the bar for acceptance.

**Justification For Why Not Higher Score:**

One could argue that a more general theory of multimodal transfer would have a higher impact.

**Justification For Why Not Lower Score:**

The paper is strong enough to be accepted, and will likely inspire a number of new works, have practical implications, etc.

---

### Decision · Program_Chairs · 2024-01-16

Accept (poster)